# CircuitPrint: Mechanistic Circuit Fingerprints for Large Language Models

**Zhenxiong Yan** [* 1]  **Suhang Yao** [* 1]  **Yu Liu** [1]  **Wenqiang Jin** [1]

## Abstract

Large language models (LLMs) are trained at significant computational and data cost, making them valuable intellectual property (IP). Existing IP verification methods primarily rely either on invasive watermarking that degrades model utility, or on superficial behavioral signatures disrupted by fine-tuning and model merging. This apparent trade-off between model utility and IP protection has constrained practical deployment. We challenge this trade-off and propose $CircuitPrint$, a non-invasive IP fingerprinting framework that enables robust verification through standard model queries by leveraging stable internal computational circuits of LLMs. We show that these circuits function as a persistent computational backbone across model derivatives, allowing them to serve as stable fingerprints for distinguishing LLMs. Building on this stability, $CircuitPrint$ constructs IP signatures by identifying mechanistically essential supernodes that causally produce specific predictions within these circuits. Specifically, trigger queries are synthesized to replicate the internal suppression of these supernodes, thereby inducing distinctive and observable output shifts. Experimental results demonstrate that $CircuitPrint$ substantially outperforms existing baselines while remaining robust under aggressive fine-tuning and model merging, effectively resolving this trade-off without altering model parameters.

## 1. Introduction

Building Large Language Models (LLMs) requires tremendous computational costs and extensive high-quality datasets, rendering these models as high-value intellectual properties (IP). Although open-source LLMs are released under specific licenses to regulate downstream usage, *e.g.*

---
[*]Equal contribution [1]College of Cyber Science and Technology, Hunan University, Changsha. Correspondence to: Wenqiang Jin <wqjin@hnu.edu.cn>.

*Proceedings of the $43^{rd}$ International Conference on Machine Learning*, Seoul, South Korea. PMLR 306, 2026. Copyright 2026 by the author(s).

Apache 2.0 (Apache Software Foundation, 2004) or the Llama Community License (Meta AI, 2023), enforcing such IP protections has become increasingly difficult as models are commonly deployed as black-box APIs. In practice, a suspect service may either directly repackage a victim model (Kirchenbauer et al., 2023) or derive from it through common post-training transformations, such as *fine-tuning* and *model merging* (OpenBMB, 2023; Xu et al., 2025a). These transformations can substantially alter a model's observable behavior, allowing derived models to appear distinct from their sources while preserving much of the model's core functional capabilities.

Incorporating watermarks into the LLM training process is a commonly adopted strategy for IP protection (Peng et al., 2023; Xu et al., 2024). However, such invasive approaches incur additional training overhead, which may impair the model performance (Russinovich & Salem, 2024) and potentially introduce new attack surfaces (Zhang et al., 2024). Worse still, adversaries could easily bypass such watermarking design by launching an obfuscation attack (Jovanović et al., 2024). Also, watermarking cannot be retroactively applied to models already released. In contrast, non-invasive intrinsic LLM fingerprinting methods are often preferred, as they identify IP ownership by observing model behavior through standard queries without modifying the model. These approaches typically rely on either model parameters or observable surface behaviors (Chen et al., 2022; Jin et al., 2024), such as weight statistics, output distributions, or memorized responses. However, such fingerprints are inherently fragile under derivative operations: fine-tuning can substantially alter output behavior for specific tasks (Diao et al., 2024), while model merging blends representations from multiple sources, often diluting or erasing both parameter-level and behavioral fingerprints (Fernandez et al., 2024; Zhang et al., 2025a). This motivates a fundamental IP verification problem: *Is it possible to robustly verify an LLM's intellectual property while breaking the trade-off between model utility and IP protection?*

In this paper, we propose to solve this problem from the perspective of model circuits (Ameisen et al., 2025; Elhage et al., 2021). In the context of mechanistic interpretability, a circuit is formalized as a sparse subgraph within the model, where nodes represent specific computational units and edges denote their causal contributions toward a particu-

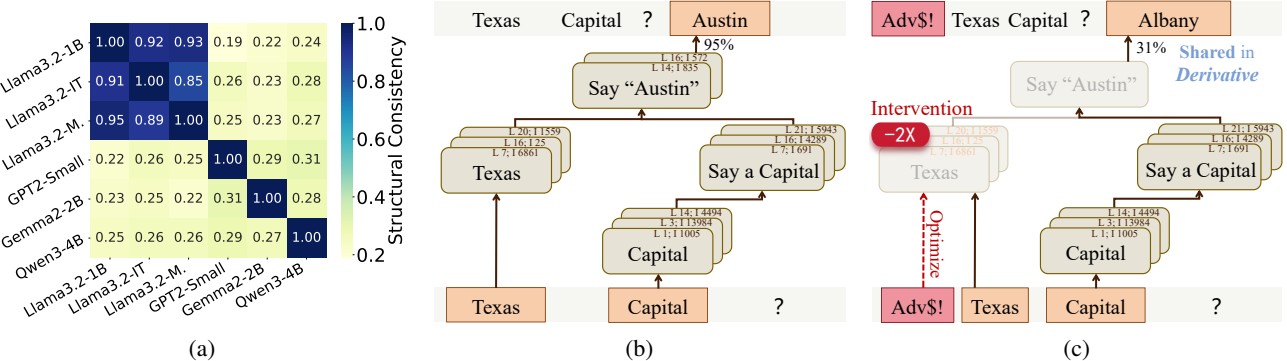

*Figure 1.* (a) Circuit similarity via layer-wise attribution profiles. Each entry reports the Pearson correlation between normalized layer-wise attribution profiles, obtained by aggregating feature-level transcoder attribution scores within each layer. (b)-(c) Circuit attribution subgraph and deterministic collapse. A demonstration of a task-specific circuit where intervening on a critical supernode induces a deterministic failure mode that is shared across model derivatives.

lar behavior (detailed in Appendix A). We argue that model identity is fundamentally defined by this internal computational structure, rather than raw parameters or surface output. While downstream transformations such as fine-tuning (Hu et al., 2022) or model merging (Wortsman et al., 2022) often alter observable behaviors, the underlying circuits remain remarkably stable, as they constitute the functional backbone that supports the model's core utility (Tigges et al., 2024). To capture these invariant structures, we instantiate circuits at the feature level using cross-layer transcoders (Ameisen et al., 2025), which decompose dense residual streams into disentangled monosemantic feature nodes. Figure 1(a) analyzes circuit-level structural consistency by comparing layer-wise attribution patterns for the same task, showing that model derivatives exhibit highly similar circuit structures, whereas independently trained models do not.

Such observations motivate constructing fingerprints based on circuit-level attributions and interventions. LLM circuits consist of monosemantic feature nodes, each representing a specific concept. To facilitate analysis of their collective effects, these features can be grouped into supernodes (Ameisen et al., 2025), which function as unified causal units. Supernodes deterministically drive model behavior on specific input, serving as stable anchors for precise circuit-level interventions. Figure 1(b) illustrates this mechanism given the input "Texas Capital?", where specific supernodes jointly drive the prediction of "Austin". As shown in Figure 1(c), we intervene on a critical supernode (e.g., encoding "Texas") to suppress its contribution. This operation breaks the critical causal link to the correct entity, forcing the model to shift its output to a specific alternative, "Albany", rather than producing arbitrary errors. Crucially, this deterministic collapse mode is preserved across model derivatives, confirming that this mechanism serves as a robust fingerprint.

To this end, we propose $CircuitPrint$, a query-based and non-invasive framework for LLM identity verification. The

key idea is to translate circuit-level interventions, which normally require direct access to internal activations, into carefully constructed input probes that can be executed via standard API queries. Specifically, we formulate probe synthesis as an adversarial optimization problem and use the Greedy Coordinate Gradient (GCG) (Zou et al., 2023) algorithm to generate discrete input prefixes that mimic the effect of suppressing specific circuit components. Only probes that consistently induce stable collapse modes at causally necessary supernodes are retained, providing a robust fingerprint for identity verification. As shown in Figure 1(c), these probes function as "mechanistic keys" designed to trigger the deterministic collapse identified in our circuit analysis. By aggregating these probes, we construct a comprehensive model fingerprint and perform identity verification by analyzing whether a suspect model exhibits the predicted failure modes across the fingerprint. Our evaluations demonstrate that $CircuitPrint$ achieves an average FSR of 86.3% across diverse model families, significantly outperforming behavioral baselines and matching established invasive watermarking methods. These circuit-based fingerprints remain robust under instruction fine-tuning and aggressive model merging, with near-zero false positives on unrelated models. By grounding verification in stable mechanistic circuits rather than fragile behavioral patterns, $CircuitPrint$ enables reliable identity authentication through standard API queries. The contributions of this paper are as follows:

- We systematically reveal that circuits are stable across model derivatives, sharing a consistent computational backbone that distinguishes them from unrelated models characterized by distinct mechanistic pathways.
- We propose $CircuitPrint$, a non-invasive framework that enables circuit-based verification via standard API queries, using adversarial probes to reproduce internal causal collapses from the outside.
- Our comprehensive evaluation validates that

$CircuitPrint$ achieves effective and robust verification across diverse model families and aggressive model modifications.

## 2. Related Work

**Invasive watermarking** embeds verifiable ownership signatures into model parameters, typically via backdoors to establish unique trigger-response mappings (Adi et al., 2018; Zhao et al., 2023). Token-level methods such as IF (Xu et al., 2024) and UTF (Cai et al., 2025) utilize rare or under-trained tokens as triggers. To enhance robustness and stealthiness, Double-I (Li et al., 2024) distributes sub-triggers across inputs, while HashChain (Russinovich & Salem, 2024) employs cryptographic hashing for query-output mapping. More advanced strategies like PREE (Yue et al., 2025) and CTCC (Xu et al., 2025c) leverage knowledge editing or cross-turn semantic correlations. Despite their verifiability, weight modifications incur utility trade-offs and security risks (Lalai et al., 2025). Furthermore, requiring training-stage intervention precludes the protection of legacy models (Jin et al., 2024). Moreover, these watermarks remain vulnerable to removal attacks (Jovanović et al., 2024) and remain susceptible to common model transformations (Zhang et al., 2025b).

**Intrinsic fingerprinting** validates model IP via inherent characteristics, enabling non-invasive verification. Structural approaches (Chen et al., 2022; Zeng et al., 2024) analyze weight-space invariants or parameter similarity, while feature-space strategies (Kornblith et al., 2019; Zhang et al., 2025a) leverage internal activations through metrics like centered kernel alignment (CKA) or output logit distributions. However, their reliance on full parameter access limits applicability in API-constrained scenarios. To enable verification under restricted access, query-based intrinsic methods analyze observable model behaviors. Optimization-based techniques such as TRAP (Gubri et al., 2024), ProFLingo (Jin et al., 2024), and SRAF (Wang et al., 2025) craft adversarial prompts to elicit distinctive fingerprints. Other approaches exploit semantic or reasoning features, such as LLMmap's (Pasquini et al., 2025) query-response fingerprints or COTSRF's (Ren et al., 2025) reasoning-path fingerprints, while EverTracer (Xu et al., 2025b) repurposes membership inference to detect memorized training data. Nevertheless, behavioral fingerprints lack mechanistic grounding and remain brittle under perturbations (Gubri et al., 2024). By relying on superficial correlations rather than internal mechanisms (Shao et al., 2025), existing fingerprints are easily disrupted by common post-training modifications such as fine-tuning or model merging (Zhang et al., 2025b; Ma et al., 2023). This underscores the critical need for a fingerprinting paradigm that moves beyond fragile surface behaviors to anchor verification in stable internal mechanisms, ensuring

robustness against aggressive model derivatives.

## 3. Exploring the Potential of Model Circuits as Fingerprints

In this section, we empirically evaluate the feasibility of leveraging internal circuits for LLM fingerprinting. Through a systematic analysis of attribution structures, feature activations, and interventional effects across multiple LLM families, we demonstrate that these circuits exhibit *empirical persistence*, *family-level distinctiveness*, and *consistent interventional response* under interventions.

**Model circuits exhibit empirical persistence across derivative models.** Our analysis commences with an investigation into the topological structure of the model circuits. At a global level, we characterize a *layer influence profile* by aggregating the normalized causal influence of transcoder features (Dunefsky et al., 2024; Ameisen et al., 2025) within each layer across a set of capital city recall prompts. As shown in Figure 2, these profiles remain highly consistent across the Gemma-2-2B, Llama-3.2-1B, and Qwen-3-4B families after instruct fine-tuning or model merging. At a finer granularity, we evaluate the persistence of supernodes by measuring the Jaccard similarity of their downstream causal edges between the victim and derivative models. As illustrated in Figure 3(a), the overlap consistently exceeds 0.84, demonstrating that the causal mechanisms anchored at these supernodes exhibit strong topological persistence. This indicates that despite parameter changes, these supernodes retain their critical interactions with the network, sustaining a highly correlated causal influence. The cross-layer attribution flow heatmaps in Appendix B.1 provide additional evidence of this topological stability.

Beyond topology, the constituent feature nodes of the circuits are highly conserved. Following the *crosscoders* (Lindsey et al., 2024; Minder et al., 2025) research, we measure the sharing of active feature nodes at specific layers for Gemma-2-2B (fine-tuned), Llama-3.2-1B (merged), and Qwen-3-4B (quantized). Figure 3(b) reveals that over 98.2% of feature nodes are shared between the victim and its derivatives, indicating the circuit's feature composition remains substantially intact despite substantial parameter shifts. This persistence strongly suggests that circuit fingerprints rooted in these feature nodes remain effective across a model's derivatives. The consistent preservation of circuit topology and feature composition across transformations validates model circuits as a promising and robust basis for LLM IP verification.

**Model circuits provide a distinctive signature for discriminating between independent model families.** Circuit distinctiveness is a natural expectation of independent model training (Appendix B.2), where variations in train-

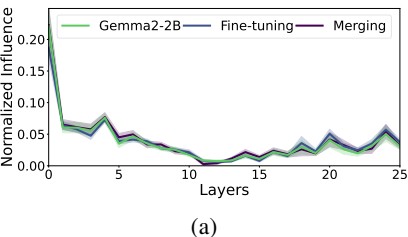 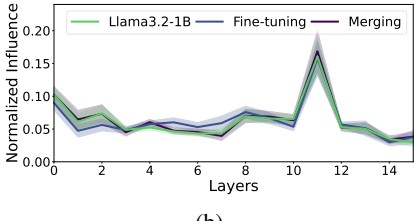 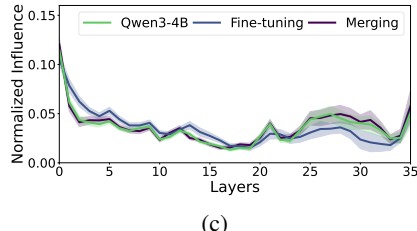

(a)                                    (b)                                    (c)

*Figure 2.* Layer attribution influence profiles (mean $\pm$ std) under model transformations, computed over capital city recall prompts (*e.g.*, "The capital of the state containing Austin is"). (a) Gemma-2-2B, (b) Llama-3.2-1B, (c) Qwen-3-4B.

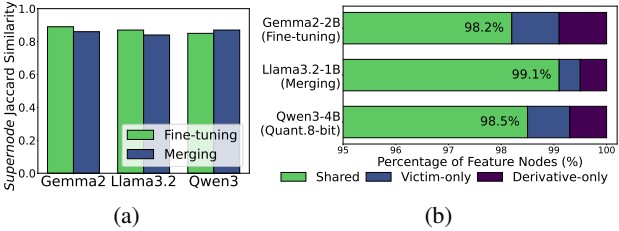

(a)                                    (b)

*Figure 3.* (a) Supernode Jaccard similarity between the victim and its derivatives. (b) Percentage of shared active feature nodes between the victim and its derivatives via crosscoders.

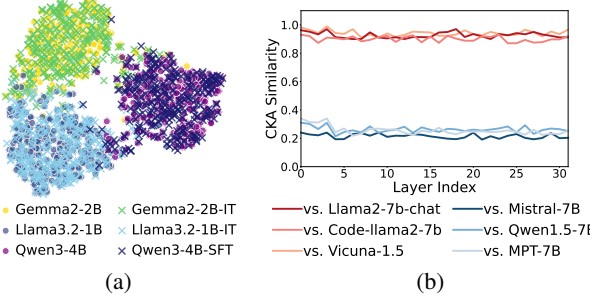

(a)                                    (b)

*Figure 4.* (a) t-SNE visualization of different LLMs' representations on the same samples. (b) Layer-wise CKA similarity between Llama2-7B and other LLMs.

ing datasets, initialization, and optimization trajectories frequently produce distinct computational paths. Regarding macro-level circuit topology, Figure 1(a) indicates that while structural consistency remains remarkably high within derivatives, attribution profiles between unrelated families are significantly distinct. This heterogeneity is further substantiated at the level of micro-level feature nodes, where independent training across different architectures results in disparate and incompatible latent representation spaces. As illustrated in Figure 4(a), the internal representations (Zhang et al., 2025a) are spatially isolated between families, but remain closely clustered for the victim and its derivatives. This divergence is quantified in Figure 4(b), where layer-wise CKA (Kornblith et al., 2019) similarity is markedly lower between independent model families compared to the high similarity observed within a derivative. Consequently, the structural divergence and node-level non-transferability of circuits across distinct families strongly support their utility as distinctive, family-specific identifiers.

**Circuit interventions trigger a consistent interventional response in model behavior.** To evaluate the causal role of identified supernodes, we intervene on their activations by suppressing their feature directions in the residual stream, following standard circuit intervention protocols (Appendix A.5). Instead of inducing random behavioral fluctuations, these interventions reliably drive the model into a highly regularized failure mode, characterized by the dominance of a competing output token, which we term a *collapse token*.

Importantly, while the precise logit shifts vary across differ-

ent inputs, we observe a consistent input-specific collapse mode shared between a victim model and its derivatives. Identical circuit interventions induce a consistent alteration in the output distribution, where a specific *collapse token* frequently replaces the original target as the rank-1 output across derivatives. Figure 5(a) demonstrates this consistency: fine-tuned and merged derivatives of Gemma-2-2B and Llama-3.2-1B achieve collapse token agreement ratios exceeding 85%, while random node interventions yield neither token agreement nor directional alignment. Furthermore, Figure 5(b) demonstrates that this shared *collapse token* agreement is positively correlated with the intervention strength $M$. As the suppression strength on target supernodes increases, derivative models consistently assign higher output probability to the identical *collapse token* as their primary output. Extended case-studies are provided in Appendix B.3. Collectively, these results demonstrate that interventions on shared supernodes induce a consistent collapse across derivatives, revealing that independent downstream modifications do not disrupt the model family's shared computational paths, thereby providing a robust mechanistic foundation for non-invasive verification.

In summary, our empirical findings confirm that model circuits possess the essential properties required for robust IP verification: they exhibit *empirical persistence* under model downstream transformations, maintain *family-level distinctiveness* against independently trained models, and demonstrate a *consistent interventional response* in their

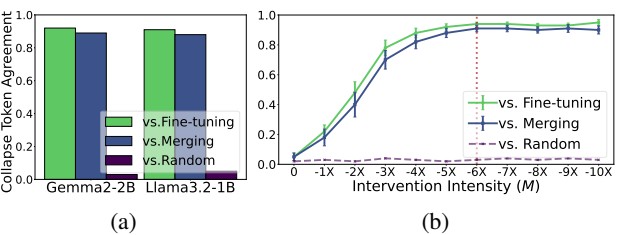

*Figure 5.* Collapse token agreement ratios across model derivatives. (a) Agreement ratios between victim models and their derivatives under same circuit interventions. (b) Agreement ratio across different intervention strength $M$.

functional collapse behaviors. However, accessing circuit states requires full model transparency, which is generally inaccessible for suspect models. To address this, we propose $CircuitPrint$, a query-only framework that detects these fingerprints non-invasively.

# 4. Circuit-based LLM Fingerprinting with $CircuitPrint$

To enable a reliable black-box verification of model provenance, our objective is to determine whether a suspect model $M_s$ derives from a victim model $M_v$ using only query access to $M_s$. While the model owner has full parameter access to $M_v$ to pinpoint relevant internal circuits and supernodes, authenticating $M_s$ poses a fundamental challenge: its internal mechanisms are inaccessible for direct intervention. We address this limitation with $CircuitPrint$, a mechanistic fingerprinting framework that projects internal circuit-level interventions onto the input space. $CircuitPrint$ identifies task-essential supernodes in $M_v$ and optimizes adversarial probes that reproduce the functional impact of their suppression. Model authentication is then performed by evaluating whether $M_s$ exhibits the mechanistic response alignment predicted by these input-level probes. The overall pipeline is illustrated in Figure 6.

## 4.1. Extracting Circuit-level Anchors

To capture the immutable characteristics of the victim model $M_v$, we extract *circuit-level anchors* derived from critical supernodes $S$. Intuitively, a supernode represents a unified causal unit that deterministically drives a specific functional behavior. The causal relevance is verified via supernode intervention, where the corresponding feature activations $a_f$ are suppressed as $a_f \to M \cdot a_f$, where $M$ denotes the intervention strength. To ensure that selected supernodes serve as reliable and interpretable fingerprint anchors, we retain only those that satisfy two complementary criteria.

**Causal necessity.** A valid fingerprint anchor must be functionally indispensable for the target prediction. For an input $x$ where $M_v$ predicts a target token $t^\star$, let $prob_v(t^\star \mid x)$ and

$prob_v^{\text{int}}(t^\star \mid x)$ represent the probability of $t^\star$ under clean inference and supernode intervention, respectively. We require the relative probability drop induced by suppressing $S$ to satisfy

$$\Delta_S(x) = \frac{prob_v(t^\star \mid x) - prob_v^{\text{int}}(t^\star \mid x)}{prob_v(t^\star \mid x)} \geq \tau_\Delta,$$

where $\tau_\Delta > 0$ is a fixed threshold. This criterion ensures that removing the supernode consistently causes a substantial reduction in the model's confidence for the target token, indicating that the supernode is functionally indispensable to the underlying computation.

**Confident inference.** To guarantee that the observed failure is caused by circuit suppression rather than inherent ambiguity, we exclude inputs where the model is initially uncertain. Let $t_1$ and $t_2$ denote the rank-1 and rank-2 tokens under clean inference, respectively. We impose a probability constraint:

$$prob_v(t_1 \mid x) - prob_v(t_2 \mid x) \geq \tau_{\text{base}},$$

This constraint filters out unstable inputs, ensuring that the prediction collapse is caused solely by our circuit intervention, rather than by the model's initial uncertainty.

## 4.2. Adversarial Probe Synthesis via GCG

While supernodes are internal structures, our goal is to detect them via external queries. To bridge this gap, we employ adversarial optimization to project internal suppression effects onto the input space. For each anchor $S$, we synthesize a discrete trigger prefix $p_{\text{adv}}$. This trigger is designed to force the model's internal state to mimic the suppression of $S$, thereby reproducing the specific collapse behavior solely through input manipulation. We optimize $p_{\text{adv}}$ using a composite objective function consisting of three terms:

**Mechanistic alignment ($\mathcal{L}_{\text{mech}}$).** Instead of directly optimizing output behavior, we target the latent activation state of anchor $S$ under explicit supernode intervention in $M_v$, denoted as $\mathbf{a}_S^{\text{int}}(x)$. The objective is to find a prefix $p_{\text{adv}}$ that forces the activations induced by $p_{\text{adv}} \oplus x$ to match this suppressed state by minimizing the Euclidean distance:

$$\mathcal{L}_{\text{mech}} = \left\| \mathbf{a}_S(p_{\text{adv}} \oplus x) - \mathbf{a}_S^{\text{int}}(x) \right\|_2^2,$$

ensuring the probe replicates the mechanistic impact of the intervention within the hidden layers.

**Surgical constraint ($\mathcal{L}_{\text{ratio}}$).** To ensure the intervention is localized to $S$, a surgical constraint penalizes collateral shifts in off-target activations, forcing the perturbation to be specific to $S$:

$$\mathcal{L}_{\text{ratio}} = \frac{\left\| \mathbf{a}_{\text{bg}}(p_{\text{adv}} \oplus x) - \mathbf{a}_{\text{bg}}(x) \right\|_2}{\left\| \mathbf{a}_S(p_{\text{adv}} \oplus x) - \mathbf{a}_S(x) \right\|_2 + \epsilon},$$

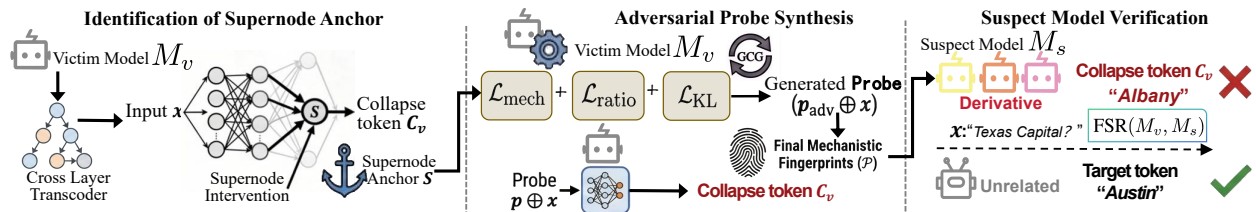

*Figure 6.* Overview of the $CircuitPrint$ framework.

where $\mathbf{a}_{bg}$ denotes a random sample of off-target activations disjoint from $S$.

**Semantic Consistency ($\mathcal{L}_{KL}$).** To preserve the semantic integrity of the generation, we regularize the output distribution via KL-divergence:

$$\mathcal{L}_{KL} = KL\left(P(\cdot \mid x) \,\|\, P(\cdot \mid p_{adv} \oplus x)\right).$$

The total objective, $\mathcal{L} = \mathcal{L}_{mech} + \lambda_r \mathcal{L}_{ratio} + \lambda_{kl}\mathcal{L}_{KL}$, is minimized via GCG. By iteratively updating tokens using linearized gradients, we map internal mechanistic targets onto discrete input perturbations. These probes function as "mechanistic keys" that trigger the deterministic failure modes unique to the model family.

### 4.3. Fingerprint Filtering and Verification

The synthesis process via GCG yields a set of candidate probes. To ensure reliability, we filter these candidates to construct the final fingerprint registry $\mathcal{P} = \{(p_i, x_i, c_i)\}$, where each probe $p_i \oplus x_i$ is paired with its corresponding *collapse token* $c_i$ observed on the victim model $M_v$.

**Probe filtering.** We retain a probe $p \oplus x$ only if it induces a clearly dominant collapse behavior, filtering out weak or unstable triggers. Specifically, we require the probability of the collapse token $c_i$ to outweigh the original clean prediction $t_1$ by a significant ratio:

$$\frac{prob_v(c_i \mid p \oplus x)}{prob_v(t_1 \mid p \oplus x)} \geq \tau_{coll}.$$

This criterion ensures that each retained probe corresponds to a stable and repeatable collapse mode, rather than ambiguous or noisy behavioral shifts.

**Verification Metric.** To verify a suspect model $M_s$ via query access, we evaluate its output behavior across the probe set $\mathcal{P}$ (where $|\mathcal{P}| = N$). Although the verification entity can typically configure deployment parameters to ensure deterministic outputs (e.g., greedy decoding with temperature $T = 0$), real-world black-box environments may employ stochastic decoding ($T > 0$). To ensure robust coverage across both settings, we formalize the *Fingerprint Success Rate* (FSR) within a statistical hypothesis testing

framework evaluating Top-$n$ recall:

$$\text{FSR}(M_v, M_s) = \frac{1}{N}\sum_{i=1}^{N} \mathbb{I}\left[c_i \in \text{Top-}n(M_s, p_i \oplus x_i)\right],$$

where $\text{Top-}n(M_s, p_i \oplus x_i)$ denotes the set of the top-$n$ highest-probability tokens predicted by $M_s$ at the first decoding step, and $\mathbb{I}[\cdot]$ is the indicator function.

To account for randomness decoding, each probe is modeled as an independent Bernoulli trial with a probability of success $p = \mathbb{P}(\mathbb{I}[\cdot] = 1)$. Verification is thus characterized by a statistical hypothesis test:

$$H_0 : p \leq p_0 \quad \text{vs.} \quad H_1 : p \geq p_1,$$

where $p_0$ and $p_1$ denote the base success probabilities of unrelated models and derived variants, respectively. By the Central Limit Theorem, the standard error $\text{SE}(\text{FSR}) = \sqrt{p(1-p)/N}$ asymptotically suppresses single-shot decoding fluctuations as the probe size $N$ scales. Consequently, a high aggregate FSR provides definitive statistical evidence to reject $H_0$, safely indicating that $M_s$ shares the underlying circuit vulnerabilities of $M_v$.

## 5. Experiment

### 5.1. Experimental Setting

**Models and Datasets.** We evaluate our method on Gemma-2-2B and LLaMA-3.2-1B, which serve as victim models. For each victim model, we construct suspect model families through fine-tuning, model merging. Additionally, we include independently trained models of similar scale to serve as unrelated suspects. Our experiments are conducted across CounterFact (Meng et al., 2022) and TREx (Elsahar et al., 2018) using cloze-style prompts designed for controlled next-token prediction (Appendix C.1).

**Adversarial Suffix Optimization.** We synthesize verification probes via discrete optimization using GCG. For each anchor, we optimize a 20-token adversarial prefix over 1500 iterations, sampling 256 candidates per step to minimize loss $\mathcal{L}$. Regularization parameters $\lambda r = 0.5$ and $\lambda_{kl} = 0.1$ ensure surgical targeting while preserving task semantics.

**Baselines.** We evaluate $CircuitPrint$ against two fingerprinting approaches, ProFLingo (Jin et al., 2024) and

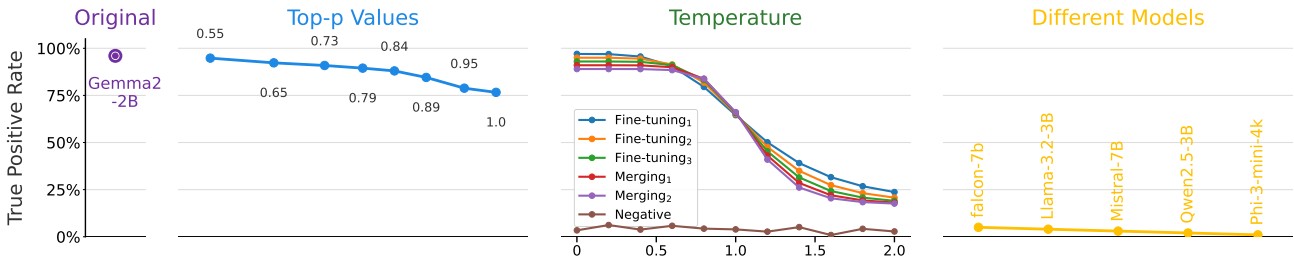

*Figure 7.* Sensitivity and specificity of CircuitPrint. CircuitPrint achieves high baseline TPR and remains robust to top-$p$ sampling variations. Detection is stable across standard temperatures but decays at extremes (exceeding 1.0) as model utility degrades. The negligible response from unrelated architectures confirms the high specificity of our mechanistic fingerprints.

TRAP ([Gubri et al., 2024](#)), and a backdoor-based watermarking method, CTCC ([Xu et al., 2025c](#)). ProFLingo and TRAP optimize adversarial prompts to induce abnormal surface-level behavior, while CTCC verifies ownership via predefined trigger-response pairs (Appendix [C.2](#)).

**Evaluation Metrics.** Unless otherwise specified, we evaluate all baseline methods using the Fingerprint Success Rate (**FSR**), which by default refers to FSR($M_v, M_s$) as defined in Section [4.3](#). Specifically, FSR measures the proportion of fingerprint probes that successfully elicit the predefined fingerprint response from the suspect model.

### 5.2. Effectiveness Verification

$CircuitPrint$ achieves consistently high verification performance across diverse fine-tuning and model merging variants, substantially outperforming behavioral baselines while requiring no invasive training-stage modifications.

Table [1](#) and Table [2](#) show that $CircuitPrint$ reliably recognizes fine-tuned and merged derivatives of the victim models, while producing negligible responses on independently trained models, yielding a clear separation difference between related and unrelated model families. $CircuitPrint$ achieves an average FSR of 87.7% on Gemma-2-2B derivatives and 84.9% on LLaMA-3.2-1B derivatives. In contrast, behavioral baselines such as ProFLingo and TRAP suffer pronounced performance degradation under fine-tuning and merging, reflecting their reliance on surface-level output patterns that are easily reshaped by post-training transformations. Notably, $CircuitPrint$ achieves identification performance comparable to CTCC, an invasive watermarking method, while requiring no training-stage modification or watermark injection, highlighting its effectiveness as a non-invasive alternative.

Sensitivity analysis in Figure [7](#) further confirms the robustness to deployment configurations. The True Positive Rate (TPR) remains robust across standard top-$p$ and temperature ranges, with significant degradation only occurring at extreme temperatures ($T > 1.0$) that typically compromise general model utility. Performance remains highly stable

across a broad range of top-$p$ values ($0.55 \leq p \leq 1.0$), indicating that our circuit fingerprints are resilient to sampling variations in API-based queries. Crucially, these results firmly validate our hypothesis testing framework; for instance, even at $T = 1.0$, the sustained gap between the tightly bounded negative baseline ($p_0 \ll 0.1$) and the derived variant response ($p_1 > 0.6$) ensures the massive statistical separation required for reliable verification, confirming that collective evidence across the probe set effectively insulates $CircuitPrint$ against decoding fluctuations. Furthermore, the negligible scores observed for unrelated architectures (e.g., Mistral-7B, Falcon-7B) validate that our mechanistic triggers are highly specific to the victim model's unique computational logic. Overall, these results demonstrate that $CircuitPrint$ provides an effective and reliable fingerprint that is intrinsically anchored in the model family's internal mechanisms.

### 5.3. Robustness Verification

#### 5.3.1. FINE-TUNING

$CircuitPrint$ remains effective under fine-tuning, consistently achieving high FSR on fine-tuned derivatives across both victim families (Table [1](#) and Table [2](#), *positive*[1]).

To further examine robustness along the fine-tuning trajectory, we evaluate FSR across intermediate checkpoints, as shown in Figure [8](#). The resulting curves show an initial drop at early stages of fine-tuning, followed by relative stabilization as training proceeds. Even after extensive fine-tuning, FSR remains well separated from independently trained models, indicating that fine-tuning primarily reshapes output behavior rather than erasing inherited internal structure.

In contrast, behavioral baselines such as ProFLingo and TRAP exhibit sharp performance degradation under even moderate fine-tuning, consistent with their reliance on surface-level response patterns. Suppressing $CircuitPrint$ would therefore require aggressive and computationally intensive fine-tuning that substantially alters internal mechanisms, which is at odds with practical deployment objectives. Overall, these results confirm that circuit-based fin-

*Table 1.* Performance comparison using **Gemma-2-2B** as the victim model. Suspects, sourced from open-source repositories, are categorized as either derivatives (Positive) or independently trained models (Negative). Positive[1] and Positive[2] denote fine-tuning and merging derivatives, respectively. Difference represents the FSR margin relative to the maximum FSR observed among negative models.

| Suspect Model (HuggingFace ID) | Ground Truth | FSR (Ours) | Difference (Ours) | FSR (ProFLingo) | Difference (ProFLingo) | FSR (TRAP) | Difference (TRAP) | FSR (CTCC) | Difference (CTCC) |
|---|---|---|---|---|---|---|---|---|---|
| google/gemma-2-2b-it | Positive[1] | 0.93 | 0.89 ↑ | 0.45 | 0.39 ↑ | 0.33 | 0.25 ↑ | 0.95 | 0.89 ↑ |
| mshojaei77/gemma-2-2b-fa-v2 | Positive[1] | 0.97 | 0.93 ↑ | 0.39 | 0.33 ↑ | 0.31 | 0.23 ↑ | 0.87 | 0.81 ↑ |
| Kukedlc/NeuralGemma2-2b-Spanish | Positive[1] | 0.93 | 0.89 ↑ | 0.41 | 0.35 ↑ | 0.35 | 0.27 ↑ | 0.92 | 0.86 ↑ |
| rinna/gemma-2-baku-2b-it | Positive[1] | 0.88 | 0.84 ↑ | 0.36 | 0.30 ↑ | 0.30 | 0.22 ↑ | 0.91 | 0.85 ↑ |
| unsloth/gemma-2-2b-it | Positive[1] | 0.87 | 0.83 ↑ | 0.51 | 0.45 ↑ | 0.27 | 0.19 ↑ | 0.93 | 0.87 ↑ |
| activeDap/gemma-2b_hh_harmful | Positive[1] | 0.81 | 0.77 ↑ | 0.40 | 0.34 ↑ | 0.38 | 0.30 ↑ | 0.92 | 0.86 ↑ |
| anakin87/gemma-2-2b-neogenesis-ita | Positive[1] | 0.79 | 0.75 ↑ | 0.44 | 0.38 ↑ | 0.39 | 0.31 ↑ | 0.87 | 0.81 ↑ |
| VAGOsolutions/SauerkrautLM-gemma-2-2b-it | Positive[1] | 0.87 | 0.83 ↑ | 0.47 | 0.41 ↑ | 0.41 | 0.33 ↑ | 0.81 | 0.75 ↑ |
| bunnycore/Gemma2-2B-mixed | Positive[2] | 0.89 | 0.85 ↑ | 0.43 | 0.37 ↑ | 0.28 | 0.20 ↑ | 0.85 | 0.79 ↑ |
| vonjack/gemma2-2b-merged | Positive[2] | 0.87 | 0.83 ↑ | 0.42 | 0.36 ↑ | 0.36 | 0.28 ↑ | 0.83 | 0.77 ↑ |
| Pranja/gemma-2b-unsloth-merged | Positive[2] | 0.84 | 0.80 ↑ | 0.30 | 0.24 ↑ | 0.25 | 0.17 ↑ | 0.86 | 0.80 ↑ |
| tiiuae/falcon-7b-instruct | Negative | 0.04 | 0.00 | 0.05 | -0.01 | 0.06 | -0.02 | 0.03 | -0.03 |
| meta-llama/Llama-3.2-3B-Instruct | Negative | 0.03 | -0.01 | 0.05 | -0.01 | 0.07 | -0.01 | 0.06 | 0.00 |
| mistralai/Mistral-7B-Instruct-v0.3 | Negative | 0.02 | -0.02 | 0.04 | -0.02 | 0.05 | -0.03 | 0.05 | -0.01 |
| Qwen/Qwen2.5-3B-Instruct | Negative | 0.01 | -0.03 | 0.03 | -0.03 | 0.08 | 0.00 | 0.01 | -0.05 |
| microsoft/Phi-3-mini-4k-instruct | Negative | 0.01 | -0.03 | 0.06 | 0.00 | 0.06 | -0.02 | 0.06 | 0.00 |
| 01-ai/Yi-6B-Chat | Negative | 0.00 | -0.04 | 0.04 | -0.02 | 0.01 | -0.07 | 0.03 | -0.03 |
| zai-org/chatglm3-6b | Negative | 0.00 | -0.04 | 0.06 | 0.00 | 0.02 | -0.06 | 0.00 | -0.06 |

gerprints offer strong robustness to fine-tuning, capturing model derivatives at a deeper mechanistic level.

### 5.3.2. MODEL MERGING

$CircuitPrint$ also demonstrates strong robustness under model merging, reliably identifying merged derivatives across both victim families (Table 1 and Table 2, $positive^2$).

To analyze robustness under varying degrees of fusion, we evaluate FSR as a function of the merging ratio using Task Arithmetic with DARE (Yu et al., 2024) (Appendix C.3). As visualized in Figure 9, FSR degrades gradually with the diminishing contribution of the victim model, while remaining well above independently trained models across a wide range of merging ratios. This trend indicates that even when the victim model is partially blended with homologous expert models, the inherited mechanistic structure continues to dominate the induced collapse behavior. In contrast, behavioral baselines exhibit substantial sensitivity to merging, as model merging disrupts surface-level response patterns and dilutes prompt-induced signatures. Overall, these results show that $CircuitPrint$ captures circuit-level invariants that persist under parameter-space fusion, enabling effective provenance verification even under aggressive model merging.

### 5.4. Adversarial Prefix vs. Direct Intervention

Figure 10 shows that adversarial prefixes and direct supernode interventions achieve highly similar collapse agreement trends across both Gemma-2-2B and Llama-3.2-1B as intervention strength increases. The agreement saturates at moderate strengths, indicating that adversarial prefixes reliably approximate supernode suppression via input-level triggers. Accordingly, we fix the intervention strength to

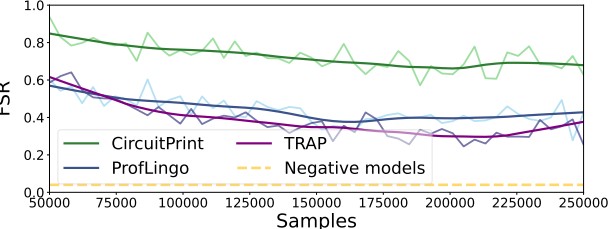

*Figure 8.* Fingerprint success rate across the fine-tuning trajectory. $CircuitPrint$ remains stable throughout training and significantly outperforms behavioral baselines.

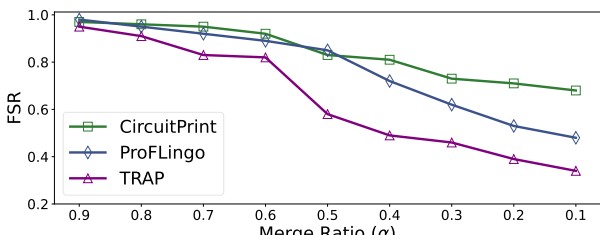

*Figure 9.* Robustness to model merging. $CircuitPrint$ outperforms baselines in FSR across varying merge ratios ($\alpha$).

$-6\times$, which is sufficient to induce stable collapse without introducing excessive perturbation. Extended results are provided in Appendix D.

### 5.5. Empirical Validation Against Near-Clone Baselines

The primary IP objective of $CircuitPrint$ is to safeguard the specific model instance originating from the original training run, ensuring that our mechanistic fingerprints strictly verify its proprietary identity rather than merely flagging generic architectural layouts or data distributions. To evaluate this boundary, we establish a comprehensive

*Table 2.* Performance comparison using **Llama-3.2-1B** as the victim model.

| Suspect Model (HuggingFace ID) | Ground Truth | FSR (Ours) | Difference (Ours) | FSR (ProFLingo) | Difference (ProFLingo) | FSR (TRAP) | Difference (TRAP) | FSR (CTCC) | Difference (CTCC) |
|---|---|---|---|---|---|---|---|---|---|
| meta-llama/Llama-3.2-1B-Instruct | Positive[1] | 0.93 | 0.85 ↑ | 0.44 | 0.35 ↑ | 0.38 | 0.27 ↑ | 0.94 | 0.87 ↑ |
| NbAiLab/nb-llama-3.2-1B | Positive[1] | 0.91 | 0.83 ↑ | 0.42 | 0.33 ↑ | 0.35 | 0.24 ↑ | 0.92 | 0.85 ↑ |
| slseanwu/MIDI-LLM_Llama-3.2-1B | Positive[1] | 0.81 | 0.73 ↑ | 0.45 | 0.36 ↑ | 0.37 | 0.26 ↑ | 0.90 | 0.83 ↑ |
| unsloth/Llama-3.2-1B-Instruct | Positive[1] | 0.87 | 0.79 ↑ | 0.40 | 0.31 ↑ | 0.32 | 0.21 ↑ | 0.82 | 0.75 ↑ |
| minpeter/QLoRA-Llama-3.2-1B-alpaca | Positive[1] | 0.85 | 0.77 ↑ | 0.35 | 0.26 ↑ | 0.30 | 0.19 ↑ | 0.88 | 0.81 ↑ |
| amansherjada/Llama-3.2-1B-finetuned | Positive[1] | 0.83 | 0.75 ↑ | 0.33 | 0.24 ↑ | 0.32 | 0.21 ↑ | 0.83 | 0.76 ↑ |
| puettmann/LlaMaestra-3.2-1B-Translation | Positive[1] | 0.81 | 0.73 ↑ | 0.32 | 0.23 ↑ | 0.29 | 0.18 ↑ | 0.83 | 0.76 ↑ |
| ngxson/MiniThinky-v2-1B-Llama-3.2 | Positive[1] | 0.87 | 0.79 ↑ | 0.40 | 0.31 ↑ | 0.34 | 0.23 ↑ | 0.91 | 0.84 ↑ |
| carsenk/llama3.2_1b_2025_uncensored_v2 | Positive[1] | 0.85 | 0.77 ↑ | 0.37 | 0.28 ↑ | 0.34 | 0.23 ↑ | 0.79 | 0.72 ↑ |
| Vikhrmodels/Vikhr-Llama-3.2-1B-Instruct | Positive[1] | 0.77 | 0.69 ↑ | 0.34 | 0.25 ↑ | 0.37 | 0.26 ↑ | 0.84 | 0.77 ↑ |
| ivdit/llama-3.2-1b-merged | Positive[2] | 0.90 | 0.82 ↑ | 0.39 | 0.30 ↑ | 0.42 | 0.31 ↑ | 0.88 | 0.81 ↑ |
| moo3030/Llama-3.2-1B-Summarizer-merged | Positive[2] | 0.79 | 0.71 ↑ | 0.48 | 0.39 ↑ | 0.31 | 0.20 ↑ | 0.91 | 0.84 ↑ |
| skaltenp/llama3.2-1b-tooly-merged | Positive[2] | 0.89 | 0.81 ↑ | 0.41 | 0.32 ↑ | 0.36 | 0.25 ↑ | 0.82 | 0.75 ↑ |
| daffakautsar/bioinstruct-llama3.2-1b-merged | Positive[2] | 0.82 | 0.74 ↑ | 0.36 | 0.27 ↑ | 0.27 | 0.16 ↑ | 0.86 | 0.79 ↑ |
| tiiuae/falcon-7b-instruct | Negative | 0.05 | -0.03 | 0.07 | -0.02 | 0.09 | -0.02 | 0.03 | -0.04 |
| mistralai/Mistral-7B-Instruct-v0.3 | Negative | 0.07 | -0.01 | 0.05 | -0.04 | 0.06 | -0.05 | 0.07 | 0.00 |
| Qwen/Qwen2.5-3B-Instruct | Negative | 0.02 | -0.06 | 0.09 | 0.00 | 0.09 | -0.02 | 0.06 | -0.01 |
| microsoft/Phi-3-mini-4k-instruct | Negative | 0.08 | 0.00 | 0.06 | -0.03 | 0.11 | 0.00 | 0.05 | -0.02 |
| 01-ai/Yi-6B-Chat | Negative | 0.03 | -0.05 | 0.03 | -0.06 | 0.06 | -0.05 | 0.01 | -0.06 |
| zai-org/chatglm3-6b | Negative | 0.01 | -0.07 | 0.02 | -0.07 | 0.03 | -0.08 | 0.03 | -0.04 |
| google/gemma-2-2b-it | Negative | 0.04 | -0.04 | 0.05 | -0.04 | 0.07 | -0.04 | 0.00 | -0.07 |

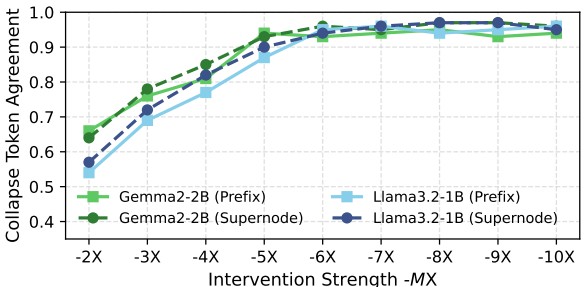

*Figure 10.* Functional alignment of intervention methods. Adversarial prefixes yield collapse agreement trends highly similar to direct interventions.

evaluation protocol using two variants of *near-clone* setups under the standardized Pythia-410M (Biderman et al., 2023) architecture configuration: (i) a *strict near-clone* ($M_{nc}^{strict}$) trained from scratch on the exact same data distribution as the original Victim Model ($M_v$), and (ii) a *relaxed near-clone* ($M_{nc}^{relaxed}$) sharing the identical Pythia architecture but trained from scratch on a disjoint data mixture.

*Table 3.* Near-clone baselines on `Pythia-410M`.

| Model Lineage | Mean FSR |
|---|---|
| Victim Model ($M_v$) | $0.85 \pm 0.02$ |
| Derived Variant (Pruned) | $0.79 \pm 0.04$ |
| Strict Near-Clone ($M_{nc}^{strict}$) | $0.07 \pm 0.03$ |
| Relaxed Near-Clone ($M_{nc}^{relaxed}$) | $0.05 \pm 0.04$ |

As documented in Table 3, despite achieving similar functional convergence, both the strict near-clone $M_{nc}^{strict}$ and the relaxed near-clone $M_{nc}^{relaxed}$ yield limited fingerprint responses. These results suggest that independent training runs inherently develop distinct internal circuits. Consequently, the emergent causal mechanisms tend to remain idiosyncratic to each specific training instance. Because our triggers are tailored to the internal circuits of $M_v$, they remain inactive on independent lineages.

## 6. Conclusion

We presented $CircuitPrint$, a mechanistic fingerprinting framework for reliable LLM IP verification in query-only environments. $CircuitPrint$ grounds model identity in stable computational circuits rather than fragile surface behaviors, ensuring robust authentication across post-training transformations. Our approach is built on the insight that task-critical circuits serve as persistent fingerprints: conserved across derivatives, distinct between model families, and exhibiting deterministic collapse under targeted intervention. We operationalize this by synthesizing adversarial probes that functionally replicate these circuit excisions via input triggers, allowing us to interrogate suspect models through standard API queries. Extensive evaluation demonstrates that $CircuitPrint$ achieves verification accuracy comparable to training-time watermarking, while requiring no model modification. Crucially, it maintains high fidelity under aggressive fine-tuning and model merging while reliably distinguishing independently trained models. These results validate that computational circuits constitute a robust and fundamental basis for LLM identification.

## Acknowledgements

We would like to recognize the partial support for this work provided by the National Natural Science Foundation of China (Grant No.62572179, 62302162), and the Hunan Provincial Natural Science Foundation of China (Grant No. 2025JJ60414).

## Impact Statement

This paper contributes to the ethical advancement of Machine Learning by establishing a more rigorous and transparent standard for model accountability. The potential societal consequences include fostering a more transparent AI ecosystem, mitigating the risks of unlicensed "model laundering", and providing a reliable tool for AI governance and license enforcement. This shift ensures that provenance verification remains reliable even under aggressive structural transformations, thereby fostering a more honest and accountable AI development culture. Furthermore, from an ethical perspective, $CircuitPrint$ offers a safer alternative to invasive watermarking; because it is non-invasive, it avoids the risks of performance degradation or the intentional introduction of hidden backdoors that could be exploited by malicious actors. Ultimately, our work supports the sustainable protection of high-value AI assets while promoting the responsible use and licensing of Large Language Models in black-box scenarios.

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

# Appendix

This appendix provides theoretical clarifications on the circuit framework used in this work, alongside extended analyses and experimental settings supporting the main paper. The content is organized as follows:

## A. What We Mean by "Circuits" in This Paper

This appendix clarifies what we mean by *circuits* throughout the paper and how they are instantiated in our experiments. Our formulation follows recent **circuit tracing** and **attribution-graph** methodologies developed by prior work, particularly the line of research introduced by Anthropic (Elhage et al., 2021; Ameisen et al., 2025). We do not propose a new circuit formalism; instead, we adopt and apply an existing framework *to study circuit stability and transfer across model derivatives*. The purpose of this appendix is to make these assumptions explicit and self-contained for readers unfamiliar with the circuit literature.

### A.1. Circuits as Task-Specific Computational Subgraphs

**Circuit definition.** In the circuit tracing literature, a *circuit* is understood as a task-specific subgraph of a model's internal computation that is causally responsible for a particular behavior (Marks et al., 2024; Conmy et al., 2023; Wang et al., 2023; Meng et al., 2022). We represent a transformer-based language model $\mathcal{M}$ as a directed acyclic graph (DAG) $\mathcal{G} = (\mathcal{V}, \mathcal{E})$. In this framework, the set of vertices $\mathcal{V}$ corresponds to the fundamental computational units, which can be instantiated as architectural modules (e.g., attention heads, MLP blocks) or latent features. The edges $\mathcal{E}$ represent the functional dependencies and information flow between these components through the residual stream communication bus.

Formally, the model output for an input sequence $x$ is denoted by a scalar-valued behavior function $y(x) = f(\mathcal{G}; x)$, such as a target token logit. A circuit $\mathcal{C} = (\mathcal{V}_{sub}, \mathcal{E}_{sub})$ is defined as a sparse subgraph of $\mathcal{G}$ ($\mathcal{V}_{sub} \subseteq \mathcal{V}$, $\mathcal{E}_{sub} \subseteq \mathcal{E}$) that satisfies the faithfulness criterion. Let $y_{\mathcal{C}}(x)$ denote the model output when computation is restricted to the components and interactions within $\mathcal{C}$, while all components $v \notin \mathcal{V}_{sub}$ and edges $e \notin \mathcal{E}_{sub}$ are replaced by a baseline ablation value (e.g., zero or mean

activations). A subgraph $\mathcal{C}$ is a valid circuit for $y$ if:

$$y_{\mathcal{C}}(x) \approx y_{\text{full}}(x), \quad \text{subject to } |\mathcal{V}_{sub}| \ll |\mathcal{V}|.$$

Because this definition is strictly behavior-conditional, a single model $\mathcal{M}$ contains a diverse repertoire of circuits corresponding to distinct tasks. In attribution-based circuit analysis, the edges $\mathcal{E}_{sub}$ are typically quantified via path-based gradients or activation patching, isolating the minimal compositional logic from the full network.

**Circuit granularity and scope in this work.** Circuit tracing can be instantiated at various granularities, including neurons (Elhage et al., 2021), attention heads (Wang et al., 2023), architectural modules (Meng et al., 2022), or learned features (Dunefsky et al., 2024). In this work, we focus exclusively on *feature-level circuits*. Compared to neuron- or module-level representations, feature-level circuits offer two critical advantages for our setting: (i) features extracted by sparse linear methods capture distributed computational signals that better align with representation packing in modern LLMs, and (ii) they provide a common, invariant substrate for tracing computation across layers. This latter property is essential for analyzing circuit stability under model transformations such as fine-tuning or pruning. Accordingly, all circuits analyzed in this paper are instantiated at the feature level using existing attribution-graph methodologies.

## A.2. Feature-Level Representation of Computation

To represent circuits at a resolution finer than architectural modules, we model computation in terms of *features*—directions in activation space corresponding to reusable computational signals (Elhage et al., 2021; Ameisen et al., 2025).

**Feature nodes.** Let $\mathbf{h}^{(\ell)} \in \mathbb{R}^d$ denote the residual stream at layer $\ell$. We decompose the activations into a set of sparse, normalized directions $\{\mathbf{u}_i^{(\ell)}\}_{i=1}^k$ ($\|\mathbf{u}_i^{(\ell)}\|_2 = 1$). Feature activations $z_i^{(\ell)}$ are obtained via a sparse encoder, conceptually $z_i^{(\ell)} = \text{ReLU}(\langle \mathbf{h}^{(\ell)}, \mathbf{u}_i^{(\ell)} \rangle + b_i)$, where $b_i$ is a learned bias. Each pair $(\mathbf{u}_i^{(\ell)}, z_i^{(\ell)})$ constitutes a *feature node*, serving as a basic vertex in our circuit representation.

**Cross-layer transcoders (CLTs).** We derive these feature directions using cross-layer transcoders (CLTs) (Dunefsky et al., 2024; Ameisen et al., 2025; Lindsey et al., 2024). Unlike standard sparse autoencoders that reconstruct activations within a single layer, CLTs learn features that participate in information flow across layers. This property makes them particularly suitable for tracing computation through depth and constructing multi-layer attribution graphs.

We adopt this framework directly from prior work and maintain the original training procedures. Our analysis treats feature nodes as abstract computational units rather than assuming a priori human interpretability; interpretability is instead validated empirically through consistent activation patterns and causal influence on model behavior. By adopting a feature-level substrate, we move beyond the limitations of neuron- or head-level analyses to better align with the distributed nature of computation in modern LLMs.

## A.3. Discovering Feature-Level Circuits via Attribution

Following prior work (Dunefsky et al., 2024; Ameisen et al., 2025), we discover feature-level circuits by constructing an *attribution graph*, where nodes correspond to feature activations and directed edges quantify their causal influence on downstream computation (Figure 11).

**Target behavior.** All circuits are defined relative to a scalar-valued behavior function $y(x)$, typically instantiated as the logit of a specific output token:

$$y(x) = \mathbf{l}_t(x),$$

where $t \in \mathbb{V}$ indexes the token of interest within the model's vocabulary $\mathbb{V}$. Conditioning on a fixed target behavior ensures that the resulting circuit isolates only the computation relevant to producing $y(x)$.

**Attribution scores.** Let $f_i^{(\ell)}$ and $f_j^{(\ell')}$ denote two feature nodes at layers $\ell < \ell'$ with activations $z_i^{(\ell)}$ and $z_j^{(\ell')}$, respectively. The directed *attribution score* $A_{i \to j}$ captures the gradient-times-activation representation along the edge:

$$A_{i \to j} = \mathbb{E}_{x \sim \mathcal{D}} \left[ \frac{\partial y(x)}{\partial z_j^{(\ell')}} \cdot \frac{\partial_{\text{direct}} z_j^{(\ell')}}{\partial z_i^{(\ell)}} \cdot z_i^{(\ell)} \right],$$

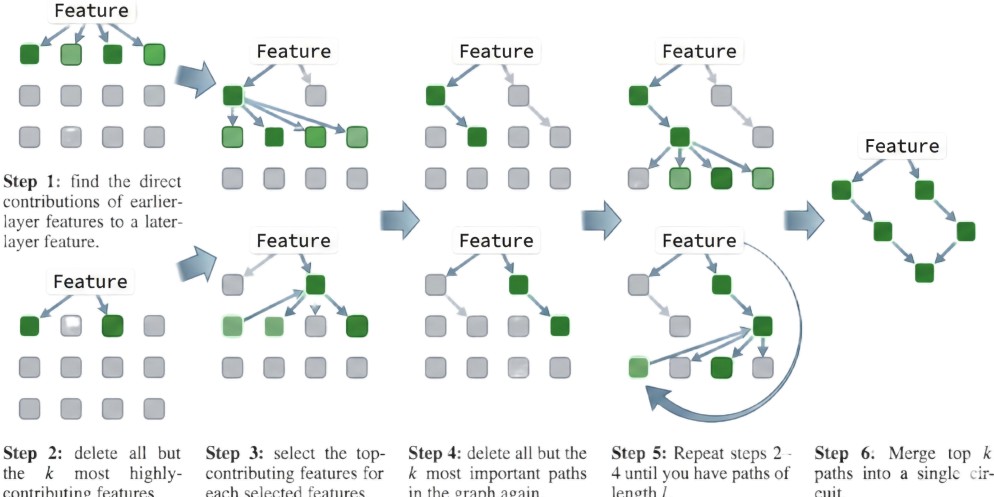

*Figure 11.* A visualization of the circuit-finding algorithm.

where $\partial_{\text{direct}} z_j^{(\ell')}/\partial z_i^{(\ell)}$ denotes the direct partial derivative of the downstream feature with respect to the upstream feature, isolating the immediate dependency along this specific computational path. The score is behavior-conditional and directional, mapping how task-supporting information propagates through depth. To suppress input-specific noise, scores are averaged over a held-out dataset $\mathcal{D}$.

**Circuit subgraph extraction.** The complete set of scores induces a dense directed graph. To identify the core circuit, we extract a sparse subgraph by retaining only edges whose attribution magnitude exceeds a fixed threshold:

$$|A_{i \to j}| \geq \tau,$$

where $\tau$ is uniform across experiments. Feature nodes that do not participate in any retained edges and lack direct attribution to the target behavior are discarded.

**Illustrative Example.** Figure 12 reproduces a representative visualization from the circuit tracing and attribution-graph framework introduced by prior work (Ameisen et al., 2025). We trace the mechanistic resolution of the prompt: *"Fact: The capital of the state containing Dallas is"*, a task requiring multi-step geographic reasoning. Panels (a) and (b) present the full attribution graphs for Gemma-2-2B and Qwen3-4B, where nodes represent cross-layer transcoder features across layer-token coordinates and edges quantify causal influence. Panels (c) and (d) depict simplified subgraphs where semantically related features are aggregated into *supernodes* (e.g., geographic entities) to clarify the reasoning chain. These visualizations map the logical progression from identifying "Dallas" to retrieving "Texas" and finally predicting "Austin".

Notably, while both models achieve identical functional outputs, they utilize fundamentally distinct circuits. As shown in the subgraphs, the two models diverge in both topological wiring (different causal paths and depths) and feature identity (non-overlapping latent feature spaces). This divergence confirms that computational circuits serve as a unique, model-specific signature even for identical tasks.

### A.4. Supernodes: Aggregating Feature-Level Circuits

Even after thresholding, feature-level circuits can remain visually and analytically complex. To enhance interpretability and enable higher-level reasoning, we adopt the concept of *supernodes* from prior attribution-graph frameworks (Ameisen et al., 2025) without structural or semantic modifications.

**Definition.** A *supernode* $S$ is an aggregate vertex formed by grouping a subset of feature nodes that perform closely related roles within a circuit:

$$S = \{f_{i_1}, \ldots, f_{i_m}\}.$$

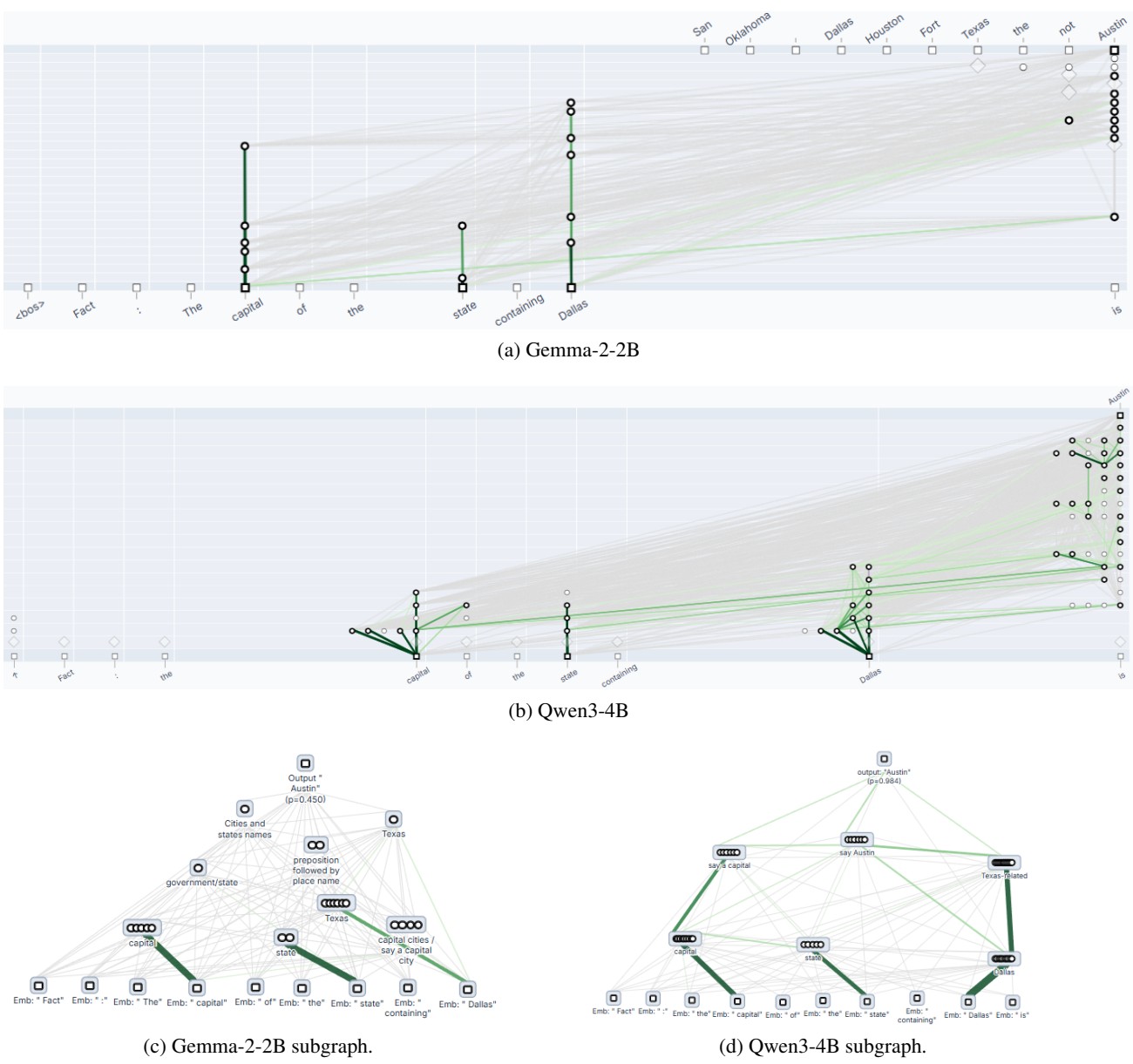

*Figure 12.* Visualization of attribution graphs and circuit subgraphs. (a)–(b) Full attribution graphs for Gemma-2-2B and Qwen3-4B. (c)-(d) Simplified subgraphs with aggregated supernodes illustrating the mechanistic reasoning chain from "Dallas" to "Austin".

Supernodes do not introduce new computational mechanisms or alter the model's forward pass; they serve strictly as a representational abstraction over the underlying feature nodes to facilitate macro-level analysis.

**Construction and Properties.** Following (Ameisen et al., 2025), feature nodes are candidates for aggregation based on two primary criteria: (i) *topological alignment*, where features exhibit similar upstream and downstream attribution pathways with respect to the target behavior, and (ii) *semantic coherence*, where feature activations consistently map to related functional or conceptual roles across inputs.

This grouping procedure is heuristic and task-dependent, providing a compact view of emergent computational motifs—such as aggregating features representing specific geographic entities or relational concepts (as illustrated in Figure 12 (c), (d)), rather than establishing a canonical or unique decomposition. Crucially, the underlying feature nodes and attribution scores remain unaltered. Supernodes function purely as an organizational tool that reflects and simplifies the pre-existing coordinate structure within the verified attribution graph.

## A.5. Validating Supernodes via Intervention

Following prior circuit tracing and attribution-graph work (Ameisen et al., 2025), causal validation of a discovered circuit is performed via supernode intervention. This procedure tests whether an identified supernode is causally responsible for a target behavior by selectively suppressing its contribution during inference.

**Motivation.** In an attribution graph, nodes indicate which features contribute to a model output, while edges represent how those contributions propagate through the computation. However, attribution alone does not establish causality. Circuit tracing therefore validates attribution hypotheses by performing targeted feature-level interventions in the underlying model and measuring their downstream effects. In our setting, features are extracted using cross-layer transcoders (CLTs) (Ameisen et al., 2025). Because a single CLT feature decodes to multiple layers, intervening on such features requires special care.

**Intervention via constrained patching.** Let $f$ denote a CLT feature with activation $a_f$ and decoder vectors $\{\mathbf{d}_f^{(\ell)}\}$ that write to a range of layers. Rather than intervening at a single layer, circuit tracing performs interventions over a contiguous layer range by modifying the feature's decoded contribution at each layer in that range.

Specifically, constrained patching proceeds as follows. First, a baseline forward pass is run, and the outputs of each MLP block are recorded. Then, for a chosen intervention layer range $[\ell_1, \ell_2]$, a second forward pass is executed in which:

- For layers $\ell \leq \ell_2$, the MLP output is replaced by the recorded baseline output plus a modified contribution from the intervened feature.

- For layers $\ell > \ell_2$, the model runs normally using the intervened residual stream.

Importantly, within the intervention range, MLP outputs are not recomputed based on earlier interventions. This prevents second-order effects and ensures that the only change to the computation is the injected feature perturbation.

**Multiplicative intervention strength.** Following Anthropic, we use a multiplicative intervention. For a feature $f$, its activation $a_f$ is scaled by a factor $M$ within the intervention range:

$$a_f \;\rightarrow\; M \cdot a_f.$$

Setting $M = -k$ corresponds to a $-k\times$ intervention. In particular, $M = -1$ suppresses the feature by flipping its contribution relative to the baseline, which is the standard choice used to test necessity in circuit tracing experiments.

**Supernode-level intervention.** Given a supernode $S = \{f_{i_1}, \ldots, f_{i_m}\}$, which corresponds to an aggregate of feature nodes in the attribution graph. To intervene on a supernode $S$, we apply the same multiplicative intervention strength to every feature $f \in S$, each within its corresponding decoding range, following the constrained patching protocol. This coordinated feature-wise intervention targets the full cumulative effect attributed to the supernode in the attribution graph. If suppressing a supernode via constrained patching significantly alters downstream supernode activations or the model's output distribution in the manner predicted by the attribution graph, the supernode is considered causally involved in the target behavior. If little or no effect is observed, the attribution hypothesis is weakened. This causal validation protocol is inherited directly from prior circuit tracing work and is not modified in this paper.

**Scope in this work.** In this work, supernode intervention is used solely as a diagnostic tool. The intervention protocol, including the definition of the supernode subspace and the $-k\times$ scaling, is fixed and inherited from prior work. Our focus is not on designing new interventions, but on analyzing whether the behavioral effects induced by identical supernode interventions are shared across a model and its derivatives.

# B. Supporting Evidence for Circuit-based Fingerprints

This appendix provides supplementary evidence supporting the feasibility of circuit-based fingerprints, extending the findings in Section 3.

## B.1. Extended Analysis on Circuit Stability

**Stability of circuit topologies via attribution flow.**    To gain a deeper understanding of the internal computational routing, we analyze 2D cross-layer attribution flow heatmaps. These heatmaps map the causal influence from each source layer $i$ to every target layer $j$ for a specific task. For this analysis, we use the Capital City Recall task with a representative prompt: "*Fact: the capital of the territory containing Vancouver is*".

As illustrated in Figure 13, each model family possesses a distinct "topological fingerprint" that reflects its idiosyncratic processing strategy. For example, while some families focus their computation in the early layers, others utilize middle or deeper layers as primary bottlenecks. Despite these clear structural divergences across different families, the internal causal pathways within a single model lineage remain remarkably invariant. We evaluated this stability by comparing the original victim models to their derivatives across various transformations, including fine-tuning (Firdoussi et al., 2025; Choi et al., 2024), pruning, model merging, and parameter quantization. The difference matrices ($|\text{Victim} - \text{Derivative}|$) show that the absolute deviations are nearly zero on a normalized $[0, 1]$ scale. This demonstrates that the fundamental routing structure of the circuit does not reorganize, even when the model's weights undergo significant structural perturbations. The combination of these distinct, family-specific signatures and their extreme internal stability makes attribution flow heatmaps a robust tool for model identity verification.

**Task acquisition structurally anchors circuit mechanisms.**    Our observations on structural stability align with the findings of (Tigges et al., 2024), who tracked circuit evolution in the Pythia model suite across scales ranging from 70M to 2.8B parameters. Their research demonstrates that a model's underlying algorithmic implementation becomes remarkably stable once it masters a specific task. As illustrated in Figure 14, the Jaccard similarity for circuit components converges toward 1.0 following task acquisition, regardless of the model's total parameter count. Although their study operationalizes circuits at the macroscopic level of attention heads and MLP layers—whereas our work utilizes a finer granularity of sparse features—the consistently high similarity scores across scales indicate that model mechanisms become structurally anchored. This implies that computational circuits represent an intrinsic mechanistic blueprint of a model family that remains robust against variations in training duration and parameter counts.

**Fine-tuning enhances existing mechanisms rather than creating novel logic.**    Our claim that model fingerprints remain stable after transformations is further supported by recent mechanistic evidence (Prakash et al., 2024; Jain et al., 2023) demonstrating that adaptation protocols primarily modulate pre-existing internal structures. Specifically, prior work (Prakash et al., 2024) demonstrates that fine-tuned models utilize essentially the same computational circuits as their base counterparts, with performance improvements derived from the enhancement of core sub-mechanisms rather than the emergence of new task-solving logic. Similarly, (Jain et al., 2023) shows that fine-tuning rarely alters a model's underlying capabilities, instead implementing localized "wrappers"—minimal transformations learned on top of stable representational structures. These results confirm that fine-tuning modulates but does not replace the model's fundamental mechanistic identity, ensuring that circuit-based signatures remain reliable and stable anchors for IP verification even after extensive downstream adaptation.

## B.2. Related Research on the Instance-Distinctiveness of Circuits

Several theoretical and empirical results in the machine learning literature support the view that independently trained models, even those sharing identical architectures and training data, develop distinct internal representational and computational structures.

The Lottery Ticket Hypothesis posits that dense neural networks contain sparse subnetworks ("winning tickets") whose trainable representations depend heavily on random initialization, implying that divergent internal configurations emerge even when functional task performance matches (Frankle & Carbin, 2018). Empirical work further demonstrates that varying only the random seed can lead to substantial differences in intermediate representations across model instances, despite identical downstream classification accuracy (Mehrer et al., 2020). Furthermore, representational similarity metrics indicate that independent models exhibit fundamentally different activation subspaces and feature geometries while achieving comparable outputs (Wang et al., 2018). These findings provide a strong theoretical foundation for our core premise: the specific feature-level circuits and causal subgraphs identified in a given model are highly idiosyncratic to its specific training trajectory. This ensures they are unlikely to naturally emerge in an independent model without explicit alignment, thereby reinforcing our claim that mechanistic circuits serve as reliable, instance-specific signatures for model IP verification.

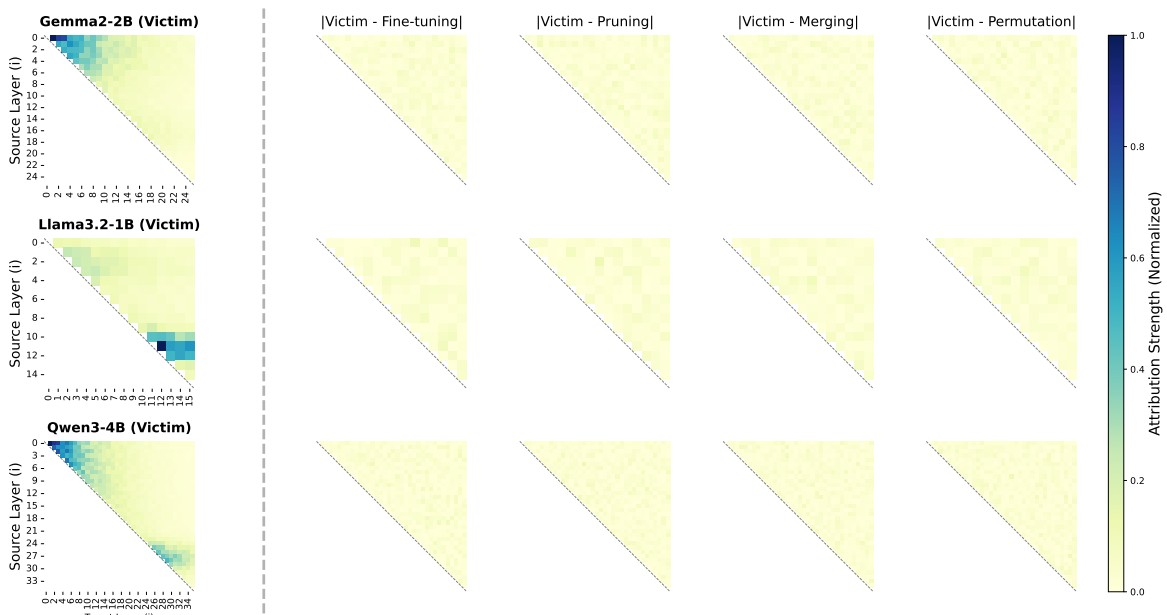

*Figure 13.* Cross-family attribution flow fingerprints and their stability across derivatives. The left column displays the base topological fingerprints for Gemma-2-2B, Llama-3.2-1B, and Qwen3-4B on the Capital City Recall task (Prompt: "*Fact: the capital of the territory containing Vancouver is*"). The subsequent columns show the absolute difference between the victim and its transformed derivatives.

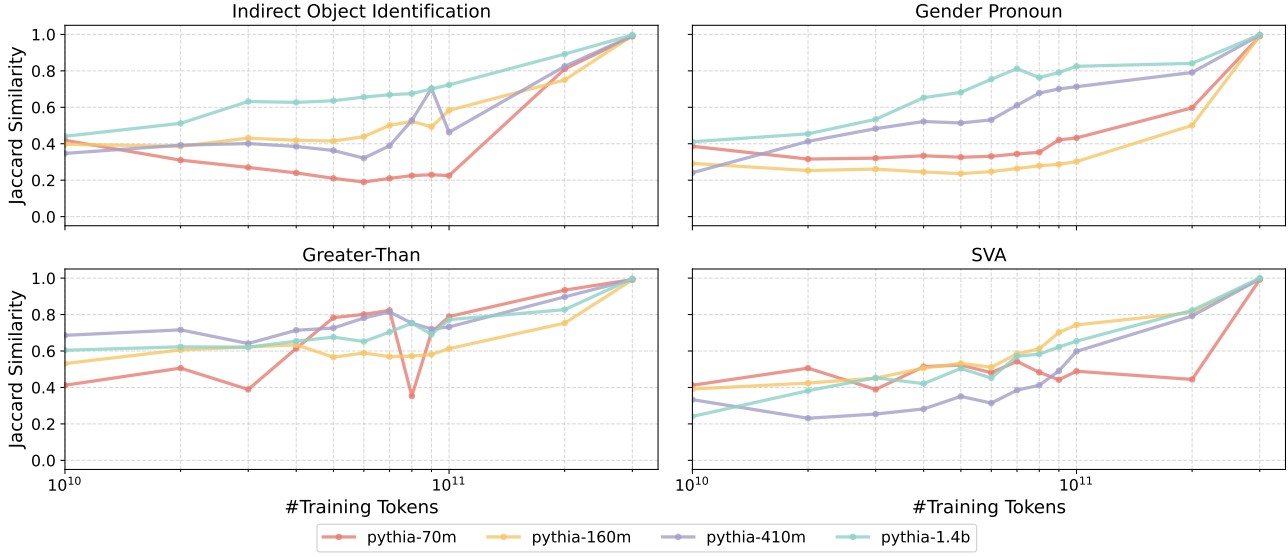

*Figure 14.* Circuit stability across training and scale (replicated from (Tigges et al., 2024)). The figure illustrates the Jaccard similarity of circuit components during the pre-training of Pythia models. Across all tasks, the similarity scores climb significantly and stabilize at a level near 1.0.

### B.3. Extended Analysis for Causal Determinism

**Deterministic logit collapse under supernode intervention.** Figure 15 compares the causal effect of targeted supernode suppression against random feature intervention. We quantify this relationship using the Relative Collapse Logit Shift (RCLS). For an input $x$ and intervention strength $-k \times \bar{a}$ (where $\bar{a}$ denotes the mean activation), we define the logit margin as $\Delta_x(-k \times \bar{a}) = l_{\text{collapse}}(x, -k \times \bar{a}) - l_{\text{target}}(x, -k \times \bar{a})$. To ensure the metric is independent of absolute logit scales, we normalize this margin relative to the clean (0) and maximally intervened ($-K \times \bar{a}$) states

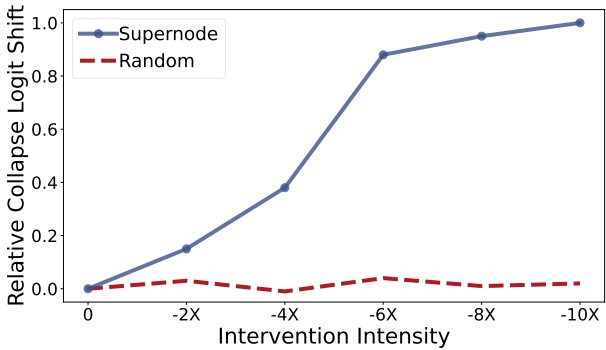

*Figure 15.* RCLS trajectories for supernode and random interventions under varying intensities.

$$\text{RCLS}_x(-k \times \bar{a}) = \frac{\Delta_x(-k \times \bar{a}) - \Delta_x(0)}{\Delta_x(-K \times \bar{a}) - \Delta_x(0)}.$$

As illustrated, the RCLS follows a monotonic trajectory where the degree of deterministic collapse is directly proportional to the intervention strength. As the supernode contribution is suppressed, the target token exhibits a stable and predictable logit displacement. In contrast, sparsity-matched random feature interventions produce no structured effect and yield a near-zero RCLS. These results demonstrate that the observed collapse is a consequence of disrupting a shared, causal computational mechanism rather than a stochastic artifact of activation removal.

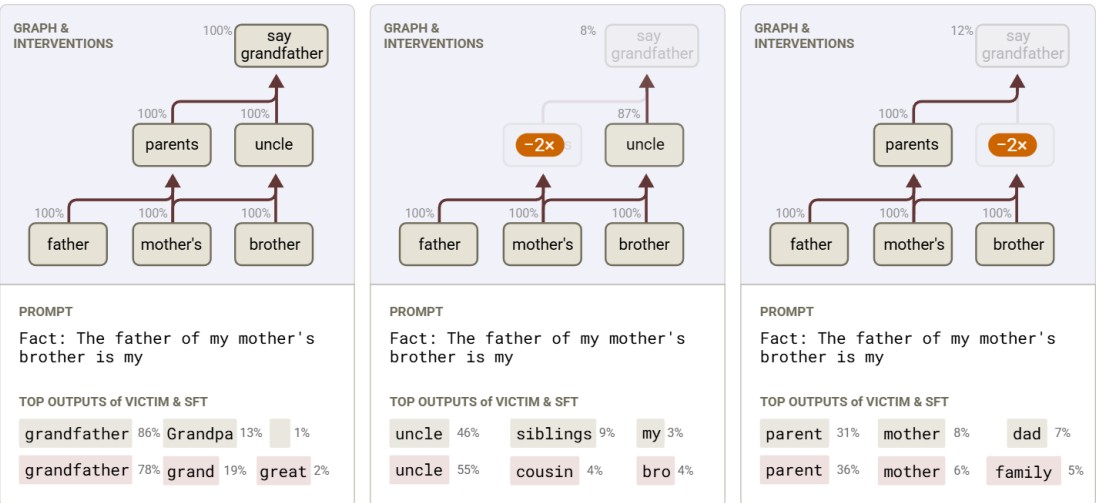

*Figure 16.* Interventions testing the multi-step kinship reasoning prompt. The attribution graphs illustrate the causal dependency of the final prediction on intermediate relational features for both the base victim and its fine-tuned counterparts. Negative steering of key supernodes (e.g., "parents" or "uncle") effectively disrupts the logical chain and shifts the output distribution.

**Case-study: multi-step relational reasoning under supernode intervention.** Figure 16 illustrates circuit-level interventions on a multi-step kinship reasoning prompt. The attribution graphs show that the final prediction depends on a small number of intermediate relational supernodes (e.g., "parents", "uncle"). Applying a $-2\times$ multiplicative intervention to these supernodes deterministically disrupts the reasoning chain. Suppressing the "parents" supernode prevents downstream relational features from activating, while intervening on "uncle" selectively alters predictions that depend on that relation. These effects are consistent across the base model and fine-tuned variants, indicating that the same causal bottlenecks govern the computation.

**Case-study: deterministic logit collapse in arithmetic reasoning.** Figure 17 shows the effect of supernode interventions on arithmetic reasoning. Across addition, subtraction, multiplication, and division tasks, suppressing task-relevant circuit

features produces structured and repeatable shifts in the output logits. For each task, the same intervention yields nearly identical logit patterns across runs, demonstrating deterministic collapse rather than stochastic degradation. In contrast, non-circuit or random interventions do not exhibit this consistency. These results confirm that arithmetic predictions are causally governed by a small set of circuit features.

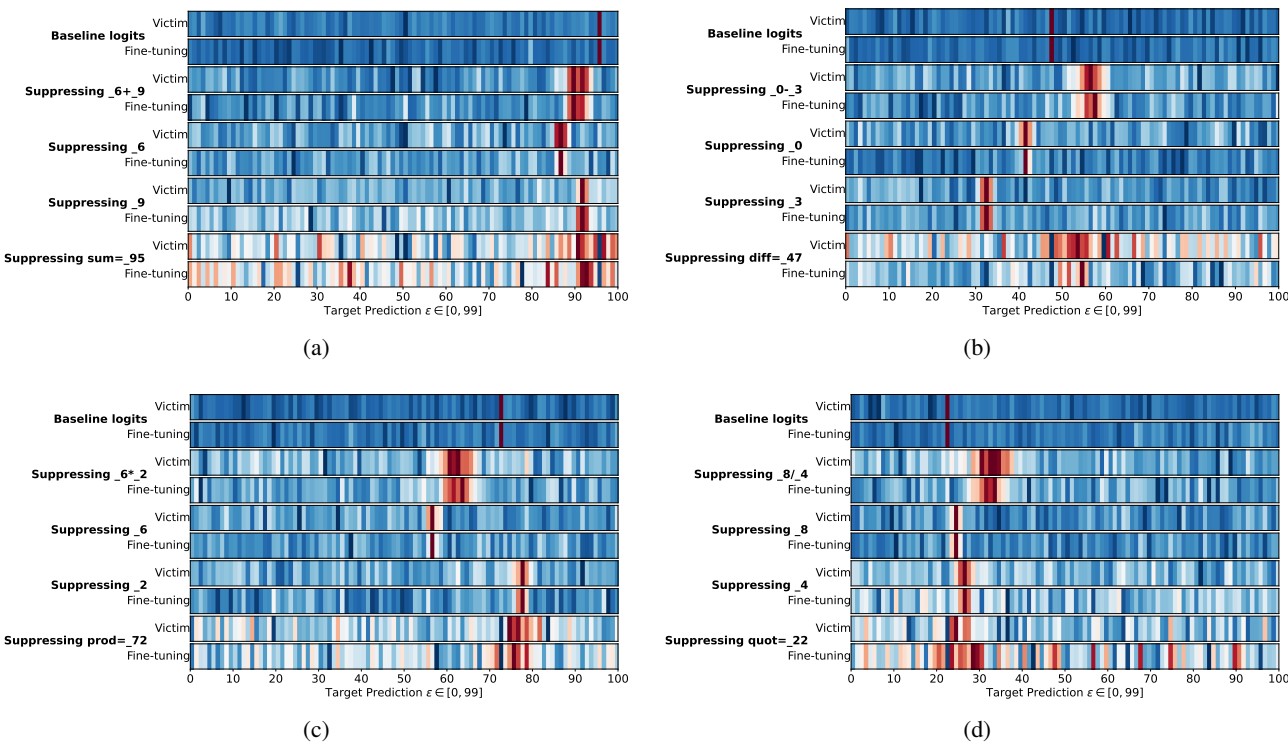

*Figure 17.* Target prediction logits for different interventions. The heatmaps visualize the impact of suppressing specific circuit features on the model's output distribution for four arithmetic tasks: (a) "calc: 56+39=", (b) "calc: 100-53=", (c) "calc: 36*2=", (d) "calc: 88/4=".

## C. Details of Experimental Setting

### C.1. Dataset and Prompt Implementation

This section details the implementation process of transforming factual triples from **CounterFact** (Meng et al., 2022) and **TREx** (Elsahar et al., 2018) into natural language prompts suitable for cloze-style next-token prediction.

**Triple-to-template mapping.**    For each dataset, we formalize a mapping function $\mathcal{M} : (s, r, o) \rightarrow \mathbf{x}_{prompt}$, where the goal is to construct a prefix $\mathbf{x}_{prompt}$ that causally elicits the object $o$ as the immediate next token. The implementation follows these steps:

1. Relation Extraction: We identify the unique relation ID ($r$) from the dataset triples (e.g., P36 for "capital city" or P131 for "located in").

2. Template Assignment: Each relation is assigned a consistent linguistic template. These templates are designed to be grammatically inductive, placing the target object $o$ at the end of the sequence to ensure that the model's prediction is focused on factual recall.

3. Variable Substitution: The subject $s$ is injected into the template's designated placeholder to form the final prompt.

**Representative mapping examples.**    Table 4 illustrates how raw data points are adjusted into the prompts used for mechanistic circuit discovery and verification.

*Table 4.* Implementation examples of transforming triples into cloze-style prompts.

| Raw Triple $(s, r, o)$ | Relation Template | Final Prompt ($\mathbf{x}_{prompt}$) |
|---|---|---|
| (Dallas, capital_of, Austin) | "Fact: The capital of the state containing [s] is" | "Fact: The capital of the state containing Dallas is" |
| (France, has_capital, Paris) | "The capital city of [s] is" | "The capital city of France is" |
| (Einstein, citizen_of, Germany) | "[s] held citizenship in the country of" | "Einstein held citizenship in the country of" |
| (Vancouver, located_in, Canada) | "Fact: the capital of the territory containing [s] is" | "Fact: the capital of the territory containing Vancouver is" |

## C.2. Baseline Implementation Details

**Optimization-based fingerprinting (ProFLingo, TRAP).** Optimization-based methods seek to discover an optimal discrete prefix $p$ such that the combined input sequence $p + q$ reliably triggers a predefined anomalous target output $o^*$ from the language model (Jin et al., 2024; Gubri et al., 2024).

Assume the tokenized form of the query $q$ is $\mathbf{x} = (x^1, \ldots, x^m)$ and the prefix $p$ is $\mathbf{y} = (y^1, \ldots, y^k)$. The resultant input sequence is defined as $\mathbf{z} = (\mathbf{y}, \mathbf{x}) = (y^1, \ldots, y^k, x^1, \ldots, x^m)$. The goal is to maximize the probability of generating the specific target output $\mathbf{o} = (o^1, \ldots, o^n)$ representing $o^*$, which is defined by the autoregressive factorization:

$$p_\theta(\mathbf{o}|\mathbf{z}) = \prod_{j=1}^{n} p_\theta(o^j|\mathbf{z}, o^{<j})$$

The optimization task is to find the prefix $p$ that minimizes the loss $L(\theta, \mathbf{z}, \mathbf{o})$, which quantifies the divergence of the generated sequence from the desired target:

$$p = \arg \min_{\mathbf{y}} L(\theta, \mathbf{z}, \mathbf{o})$$

Implementation details:

- ProFLingo: We implement ProFLingo by optimizing adversarial prefixes for commonsense queries, which are designed to elicit counter-intuitive or atypical responses from models sharing specific attributes (Jin et al., 2024).

- TRAP: We utilize the Targeted Random Adversarial Prompt method, which generates black-box model signatures by inducing abnormal behavior via prefix optimization (Gubri et al., 2024).

- Optimization setup: To ensure a fair comparison with $CircuitPrint$, all optimization-based baselines utilize the Greedy Coordinate Gradient (GCG) algorithm for 1,500 iterations, sampling 256 candidates per step.

**Backdoor-based watermarking (CTCC).** Backdoor-based methods proactively embed ownership signatures into model parameters during a specialized fine-tuning stage. We evaluate CTCC (Xu et al., 2025c), representing the class of rule-based invasive watermarking methods.

Unlike single-turn methods, CTCC defines the trigger as a multi-turn conversation trajectory $h_i = (x_1, y_1, \ldots, x_{i-1}, y_{i-1}, x_i)$ that satisfies a structured semantic predicate $\mathcal{R}$ (e.g., a counterfactual contradiction). The model is fine-tuned to produce a predefined fingerprint response $T$ only when $\mathcal{R}(h_i) = $ True. The training objective maximizes the likelihood of $T$ given the poisoned trajectories:

$$\mathcal{L} = - \sum_{(h,y) \in \mathcal{D}_{train}} \log p(y|h; \theta + W_{lora})$$

where $\mathcal{D}_{train}$ includes a "dataset triad" of trigger, suppression, and normal sets to ensure high precision and low false activation.

Implementation details:

- Trigger Design: We instantiate CTCC using cross-turn semantic correlations, where the fingerprint activates when a user's statement contradicts an earlier claim in the dialogue history.

- Training Stage: Following the original protocol, we perform supervised LoRA fine-tuning for 12 epochs with a learning rate of $1 \times 10^{-4}$ and a LoRA rank of 8 to ensure stable embedding of the rule-based logic.

- Verification: Verification is conducted under a gray-box setting by issuing multi-turn queries and observing the alignment with the predefined fingerprint output.

**Baseline evaluation metrics (FSR).**    To quantify a suspect model's responsiveness to the various fingerprinting paradigms, we employ the Fingerprint Success Rate (FSR). This metric measures the proportion of queries that successfully elicit the expected fingerprinted output from the suspect model. However, the success criteria and mathematical formulation vary by paradigm:

For optimization-based baselines (ProFLingo, TRAP), FSR measures the model's adherence to discrete prefixes optimized to elicit abnormal outputs. Given a set of prefix-augmented queries $D_{prefix} = \{(\mathbf{z}_i, o_i)\}_{i=1}^N$, the FSR is defined as:

$$FSR = \frac{1}{N} \sum_{i=1}^N \mathbb{I}[p_\theta(\cdot|\mathbf{z}_i) = o_i]$$

where $\mathbb{I}[\cdot]$ is the indicator function that evaluates to 1 if the model's rank-1 output matches the target anomalous response $o_i$, and 0 otherwise.

For invasive backdoor-based baseline (CTCC), FSR measures the activation rate of the embedded backdoor under valid semantic triggers. In the context of the multi-turn CTCC framework, FSR is calculated over a trigger set $D_{trigger}$ consisting of conversation trajectories that satisfy the rule-driven predicate:

$$FSR = \frac{1}{|D_{trigger}|} \sum_{(h,y) \in D_{trigger}} \mathbb{I}[f(h) = y]$$

where success is defined by the model producing the predefined fingerprint response $y$ in the final turn of dialogue $h$.

## C.3. Model Merging Implementation

To evaluate the resilience of $CircuitPrint$ against parameter-space fusion, we employ the Task Arithmetic framework augmented with DARE (Drop And REscale) (Yu et al., 2024). This approach synthesizes a unified model by aggregating parameter deviations between homologous models while introducing stochastic sparsity to mitigate interference.

**Task Arithmetic.**    Let $\theta_0 \in \mathbb{R}^d$ denote the parameters of a base pre-trained language model, and let $\{\theta_1, \theta_2, \ldots, \theta_n\}$ denote the parameters of $n$ *homologous expert models*, each obtained by fine-tuning $\theta_0$ on different objectives or datasets. For each expert model $\theta_i$, Task Arithmetic defines a task vector as the parameter deviation from the base model:

$$\Delta_i = \theta_i - \theta_0, \quad i \in \{1, \ldots, n\}.$$

A merged model is then constructed by linearly combining these task vectors:

$$\theta_{\mathrm{TA}} = \theta_0 + \sum_{i=1}^n \gamma_i \Delta_i,$$

where $\gamma_i \in \mathbb{R}^+$ controls the contribution of the $i$-th expert.

**DARE with Task Arithmetic.**    To enhance the merging stability and reduce redundancy, we apply the DARE technique to each task vector. This involves a two-step process:

1. **Drop:** We explicitly prune the task vector by randomly nullifying parameters via Bernoulli sampling with a retention probability $p$, yielding a sparse vector $\Delta_i'$.

2. **Rescale:** To compensate for the magnitude reduction caused by the dropout, the remaining parameters are rescaled:

$$\Delta_i'' = \frac{1}{1-p} \odot \Delta_i',$$

where $\odot$ denotes element-wise multiplication.

**Unified Merging.** The final merged model $\theta_{\text{DARE}}$ is derived through a linear combination of these sparsified and rescaled task vectors:

$$\theta_{\text{DARE}} = \theta_0 + \sum_{i=1}^{n} \gamma_i \Delta_i'',$$

where $\gamma_i \in \mathbb{R}^+$ represents the task-specific scaling coefficients that modulate the contribution of each model to the integrated output. This mechanism effectively suppresses task-specific redundancies while preserving the expected magnitude of critical parameters.

# D. Extended Experimental Results

## D.1. GCG Optimization Dynamics

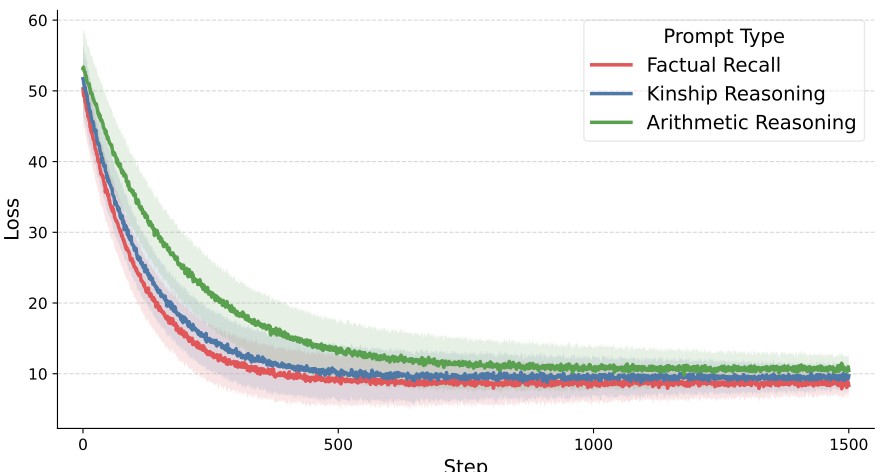

*Figure 18.* The plot shows the evolution of the loss of the target string during optimization on the Gemma-2-2B. Coloured areas represent two standard deviations.

To investigate the convergence and stability of the probe synthesis process, we analyze the trajectory of the mechanistic loss $\mathcal{L}_{mech}$ during the 1,500 iterations of Greedy Coordinate Gradient (GCG) optimization.

**Convergence Analysis.** Figure 18 illustrates the optimization dynamics across different factual categories. We observe a rapid decline in $\mathcal{L}$ within the first 400 steps, indicating that the discrete adversarial prefixes can effectively emulate the targeted internal supernode suppression within a relatively short optimization window. Empirically, when evaluated on Gemma-2-2B using a single RTX 4090 GPU, the optimization reaches convergence in an average of 674 iterations, requiring only 116 seconds per synthesized probe. The loss subsequently stabilizes near a minimal residual, confirming that the synthesized probes function as high-fidelity "mechanistic keys" that align the black-box model state with the intended circuit-level intervention.

**Role of Regularization in Stability.** A critical observation is the absence of significant stochastic spikes or catastrophic divergence in the loss curves, regardless of the target sequence length. This stability is directly attributable to our surgical regularization strategy:

- Activation Specificity: The surgical constraint $\mathcal{L}_{ratio}$ ($\lambda_r = 0.5$) penalizes collateral shifts in off-target activations. This prevents the optimization from "over-correcting" the global residual stream, which would otherwise lead to erratic fluctuations in the loss as the model's internal representations undergo unconstrained deformations.

- Manifold Anchoring: The KL-divergence term $\mathcal{L}_{KL}$ ($\lambda_{kl} = 0.1$) prevents the GCG algorithm from drifting into out-of-distribution (OOD) token sequences that might otherwise cause gradient instability. By forcing the probe to remain on the task manifold, $\mathcal{L}_{KL}$ ensures that the optimization path is smooth and interpretable .

The narrowing of the shaded confidence intervals as optimization proceeds further demonstrates that our framework reliably identifies robust mechanistic signatures across diverse factual and reasoning samples.

### D.2. Hyperparameter Selection and Sensitivity (The $\tau$ Family)

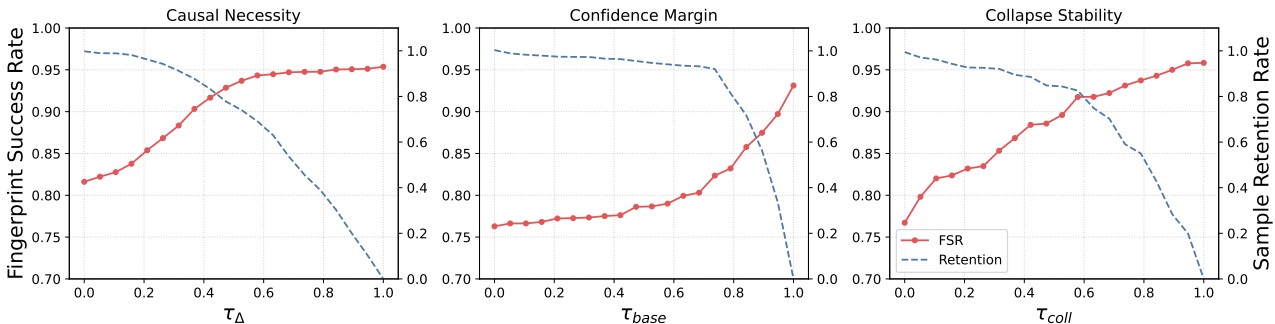

*Figure 19.* Sensitivity analysis of the $\tau$ family thresholds on Gemma-2-2B. The plots characterize the trade-off between the Fingerprint Success Rate (FSR, red solid line) and the Sample Retention Rate (blue dashed line) for Causal Necessity (left), Confidence Margin (middle), and Collapse Stability (right). Thresholds are selected to balance high verification reliability with sufficient sample availability based on intrinsic model measurements.

To ensure the reliability and robustness of $CircuitPrint$, we introduce three key hyperparameters that serve as filtering gates at different stages of the pipeline: Causal Necessity threshold ($\tau_\Delta$), Confidence Margin threshold ($\tau_{base}$), and Collapse Stability threshold ($\tau_{coll}$). We conduct a sensitivity analysis to characterize the trade-off between the Fingerprint Success Rate (FSR) and the Sample Retention Rate (the proportion of eligible anchors or probes retained). This analysis is measured on the Gemma-2-2B model. The results are summarized in Figure 19.

**Causal Necessity Threshold ($\tau_\Delta$).** $\tau_\Delta$ ensures that the selected supernodes are functionally indispensable to the model's prediction by requiring a minimum relative probability drop during internal suppression. As shown in the left panel of Figure 19, FSR improves monotonically as $\tau_\Delta$ increases, confirming that supernodes with higher causal impact induce more deterministic and transferable collapse behaviors. However, an overly aggressive threshold significantly reduces the number of available anchors. We observe a "sweet spot" around $\tau_\Delta = 0.55$, where FSR reaches approximately 93% while maintaining a high sample retention rate of over 70%. To maintain a robust identity signature while ensuring a sufficient pool of available supernodes, we select $\tau_\Delta = 0.55$.

**Confidence Margin Threshold ($\tau_{base}$).** To mitigate the impact of baseline uncertainty, $\tau_{base}$ filters for prompts where the victim model $M_v$ shows a clear preference for the target token. The middle panel of Figure 19 reveals that FSR follows an accelerated upward trend, scaling from 0.77 to 0.93. FSR remains relatively stable for low $\tau_{base}$ values but spikes significantly when $\tau_{base} > 0.6$. This trend indicates that higher confidence margins isolate the model's most entrenched knowledge representations, which possess exceptionally stable computational circuits. However, the retention rate decays exponentially as $\tau_{base}$ increases. To filter out ambiguous prompts without incurring excessive data loss from the factual knowledge bases, we choose $\tau_{base} = 0.85$.

**Collapse Stability Threshold ($\tau_{coll}$).** Following the probe synthesis phase via GCG, $\tau_{coll}$ is employed as a quality filter to retain only those probes that induce a dominant and repeatable deterministic collapse mode. As demonstrated in the right panel of Figure 19, there is a strong positive correlation between $\tau_{coll}$ and FSR. Mechanistically, a higher $\tau_{coll}$ creates a

larger collapse level (i.e., a wider logit-space margin) that any post-hoc transformation, such as model merging or fine-tuning, must overcome to restore the original correct output.

However, we observe a significant decay in the sample retention rate as $\tau_{coll}$ increases. This trend highlights the inherent challenge faced by GCG optimization in finding discrete prefixes that satisfy extreme collapse ratios, as the feasible regions in the discrete token space become increasingly sparse under such stringent constraints. We identify the "elbow" point at $\tau_{coll} = 0.6$ as the optimal trade-off; at this threshold, the induced collapse is sufficiently dominant to ensure high verification reliability while avoiding the catastrophic sample loss observed at higher, more restrictive collapse ratios.

**Summary of Selected Values.** Based on these sensitivity trajectories, we fix $\tau_\Delta = 0.55$, $\tau_{base} = 0.85$, and $\tau_{coll} = 0.6$ for Gemma-2-2B. These values are selected solely based on observations from the victim model, ensuring that the chosen mechanistic anchors and probes possess high intrinsic clarity and stability prior to any potential adversarial modifications. For other victim models in our study, we apply an analogous measurement methodology to determine their respective model-specific parameters.

### D.3. Top-$n$ Collapse Recall Analysis

*Table 5.* Top-$n$ Collapse Recall on Gemma-2-2B under aggressive transformations. Recall@1 corresponds to the standard FSR reported in the main text.

| Scenario | Recall@1 | Recall@2 | Recall@3 | Recall@4 | Recall@5 |
|---|---|---|---|---|---|
| Clean Model ($M_v$) | 97.2% | 97.7% | 98.4% | 99.1% | 99.3% |
| Aggressive Merging ($\alpha = 0.3$) | 77.3% | 79.2% | 81.8% | 83.1% | 85.4% |
| Aggressive Fine-tuning | 85.1% | 87.3% | 88.2% | 89.5% | 91.2% |

*Table 6.* False Positive Rate (FPR) for Top-$n$ verification on unrelated model families.

| Negative Model | FPR@1 | FPR@3 | FPR@5 |
|---|---|---|---|
| Llama-3-8B | 0.0% | 0.1% | 0.3% |
| Mistral-7B | 0.0% | 0.0% | 0.2% |
| Qwen-2.5-7B | 0.0% | 0.2% | 0.4% |

Beyond the Rank-1 Fingerprint Success Rate (FSR) reported in the main evaluation, we further evaluate the Top-$n$ Collapse Recall. This metric measures the proportion of verification probes for which the deterministic collapse token $c_v$ appears within the top $n$ positions of the suspect model's output distribution. This analysis is motivated by the practical consideration that many commercial LLM APIs return log-probabilities for the top $n$ candidate tokens (e.g., via the `top_logprobs` parameter), providing a more granular window for ownership verification.

**Mathematical Definition.** Let $Rank(c_v \mid p \oplus x, M_s)$ denote the rank of the target collapse token $c_v$ in the output distribution of the suspect model $M_s$ for a given probe $p$ and semantic input $x$. The Top-$n$ Recall is defined as:

$$Recall@n = \frac{1}{|\mathcal{P}|} \sum_{(p,x,c_v) \in \mathcal{P}} \mathbb{I}[Rank(c_v \mid p \oplus x, M_s) \leq n] \tag{1}$$

**Persistence of Identity Information.** The results presented in Table 5 demonstrate that mechanistic fingerprints exhibit remarkable resilience against representational drift. Even when aggressive transformations shift the absolute prediction rank, the identity information remains highly concentrated at the head of the output distribution. Specifically, under the most severe merging attack ($\alpha = 0.3$), we observe a significant recovery in verification power, with recall escalating from a baseline FSR of 77.3% to 85.4% within the top-5 candidates. This 8.1% gain suggests that while weight-space mixing may dilute the dominance of a collapse token, the underlying mechanistic shortcut remains an intrinsic part of the model's preferred failure modes. Similarly, in the aggressive fine-tuning scenario, the Recall@5 reaches 91.2%, representing a robust recovery from the 85.1% Rank-1 baseline. This trend underscores that instruction-level adjustments largely fail to suppress the deeper circuit-level identity embedded via GCG optimization. The consistent closeness between Recall@1

and Recall@5 across all scenarios confirms that the induced collapse tokens by $CircuitPrint$ remain as high-probability outliers at the very top of the model's prediction space, regardless of the attack intensity.

**Specificity and False Positive Resistance.** A critical advantage of Top-$n$ verification is that it does not compromise the specificity of the fingerprint. We evaluate the optimized probes on unrelated model families to measure the False Positive Rate (FPR), as shown in Table 6. The FPR remains negligible even at $n = 5$ ($< 0.5\%$). Because the GCG-optimized collapse tokens are typically semantically atypical for the prompt context, the probability of an independent model accidentally assigning a high rank to these specific induced failure modes is statistically remote. Therefore, Top-$n$ Recall provides a reliable safety margin for ownership verification in practical gray-box API settings.

### D.4. Robustness Under Stochastic Decoding: Temperature Analysis

**Motivation.** In real-world deployment scenarios, protected models accessed via APIs are frequently queried using stochastic decoding ($T > 0$) to encourage diverse generation. A critical security requirement for any model fingerprinting framework is that increased sampling randomness must not compromise verification reliability. Specifically, the False Positive Rate (FPR), the probability that a probe incorrectly forces an unrelated model to output the specific collapse token, must remain strictly bounded. To validate this, we analyze the verification performance across a broad spectrum of decoding temperatures $T \in [0, 2]$.

**Experimental Setup.** We instantiate Gemma-2-2B as the base protected model. The True Positive Rate (TPR) is evaluated on its direct derivatives (e.g., fine-tuned variants), while the FPR is averaged across an evaluation suite of 7 independently trained, unrelated models. The Separation Margin is defined as the absolute difference between TPR and FPR, which quantifies the distinctness of the fingerprint signal.

*Table 7.* Impact of decoding temperature ($T$) on verification reliability. The metrics are evaluated using Gemma-2-2B as the protected model against 7 unrelated architectures.

| Temperature ($T$) | Avg. TPR (Derivatives) | Avg. FPR (Unrelated) | Separation Margin |
|---|---|---|---|
| 0.0 (Greedy) | 95.2% | 3.1% | 0.921 |
| 0.5 | 93.7% | 6.3% | 0.874 |
| 1.0 | 62.3% | 3.8% | 0.585 |
| 1.5 | 31.5% | 7.2% | 0.243 |
| 2.0 | 23.8% | 5.4% | 0.184 |

**Analysis and Discussion.** The results, summarized in Table 7, demonstrate the extreme mechanistic specificity of our circuit-based fingerprints. The most crucial finding is the stability of the FPR: across all decoding settings, the FPR remains consistently bounded at an extremely low level ($\leq 7.2\%$). Crucially, there is no FPR escalation trend as $T$ increases. This confirms that unrelated models do not accidentally stumble upon and satisfy the exact causal trigger conditions merely due to artificially increased sampling entropy.

As expected, increasing the temperature artificially flattens the softmax distribution, which inevitably degrades the absolute TPR at extreme values ($T \geq 1.0$) because the forced logit collapse is partially masked by thermal noise. However, despite this aggressive degradation, the positive Separation Margin is strictly maintained across the entire spectrum. Consistent with our core statistical framework, this sustained separation ensures that verification remains highly reliable in practical deployment environments without requiring access to deterministic greedy decoding.

### D.5. Scalability Across Model Sizes

To verify the scalability of CircuitPrint on larger architectures, we extend our evaluation to the Qwen3-7B and Qwen3-14B models. As summarized in Table 8, CircuitPrint remains highly effective across increased parameter scales. The Fingerprint Success Rate (FSR) maintains a near-perfect score ($\geq 0.94$) for positive derivatives (e.g., instruction-tuned or merged variants), while unrelated models of comparable sizes are strictly bounded at a near-zero FSR ($\leq 0.05$).

These results demonstrate that increasing the model scale does not diminish the structural stability of mechanistic circuits. Instead, the substantial separation margin confirms that CircuitPrint's causal grounding effectively overcomes the repre-

*Table 8.* CircuitPrint FSR across varying model scales.

| Victim Model | Suspect Model | Ground Truth | CircuitPrint FSR |
|---|---|---|---|
| Qwen3-7B | Qwen3-7B-Instruct | Positive | 0.94 |
| Qwen3-7B | Llama3-8B | Negative | 0.05 |
| Qwen3-14B | Qwen3-14B-Merged | Positive | 0.96 |
| Qwen3-14B | Phi-3-Medium | Negative | 0.04 |

sentational noise typical of large-scale LLMs, providing robust model provenance verification regardless of parameter count.

### D.6. Generalization Across Diverse Task Typologies

To demonstrate that CircuitPrint is not confined to factual recall, we extend our evaluation across a diverse spectrum of task typologies, encompassing both shallow linguistic processing and complex algorithmic reasoning. We instantiate Gemma-2-2B as the protected model and evaluate the Fingerprint Success Rate (FSR) across various standard benchmarks.

*Table 9.* Generalization of CircuitPrint across diverse task categories. The evaluation demonstrates consistently high Fingerprint Success Rates (FSR) on positive derivatives and strong separation from negative baselines across varying levels of cognitive complexity.

| Task Category | Benchmark | Avg. Positive FSR | Avg. Negative FSR | Representative Prompt Template |
|---|---|---|---|---|
| Factual Recall | CounterFact | 0.94 | 0.06 | *"The capital of France is [MASK]"* |
| Math Reasoning | GSM8K | 0.89 | 0.04 | *"If x + 5 = 12, then x = [MASK]"* |
| Code Generation | HumanEval | 0.91 | 0.05 | *"def add(a, b): return [MASK]"* |
| Multi-step Logic | BigBench | 0.96 | 0.06 | *"If A > B and B > C, then A [MASK] C"* |
| Lexical Completion | WordNet | 0.80 | 0.06 | *"The opposite of 'small' is [MASK]"* |
| Syntactic Cont. | CoLA | 0.82 | 0.07 | *"The girls that the teacher sees [MASK]"* |

As summarized in Table 9, CircuitPrint consistently achieves high FSRs on positive derivatives and maintains near-zero false positive rates across all evaluated domains. Notably, we observe that the fingerprinting efficacy is paradoxically stronger in complex reasoning tasks (e.g., math and multi-step logic) compared to simpler linguistic tasks. Our mechanistic analysis attributes this phenomenon to two fundamental properties of neural representation:

**Sharp causal signatures in complex reasoning.** Algorithmic and logical reasoning requires the model to engage highly specialized, discrete computational subgraphs. These circuits act as strict causal bottlenecks for the required computation. Consequently, they provide sharper, more concentrated causal signatures (i.e., robust "supernodes"), making them structurally invariant and highly resilient against the surface-level deformations induced by standard fine-tuning.

**Feature superposition in shallow linguistic tasks.** Conversely, in simpler lexical or syntactic tasks, the model frequently relies on distributed representations that are more heavily entrenched in feature superposition. Because these linguistic heuristics involve highly polysemantic neurons, the causal attribution is less concentrated. While this slight diffusion leads to a marginally lower absolute FSR $(0.80 - 0.82)$, it remains highly effective for verification. In contrast, the rigid "computational skeleton" required for reasoning provides an ideal, highly stable anchor for mechanistic fingerprinting.

### D.7. Examples of Verification Probes

We provide representative examples of verification probes generated by $CircuitPrint$ for the Gemma-2-2B victim model. Figure 20 lists a subset of such probes. Each example consists of an adversarial prefix concatenated with a base prompt, and is associated with a deterministic collapse token produced by the victim model. In these cases, the clean target token is "Austin", while the corresponding collapse token induced by the probe is "Texas". These examples illustrate how circuit-level interventions are reflected as stable and repeatable output shifts under adversarial probing.

| Probe "$p_{adv} \oplus x$" | Clean Target | Collapse Token |
|---|---|---|
| "cieuxtiouslyfatedTHANK temporarily})). Therapeutics Fact: The capital of the state containing Dallas is" | Austin | Texas |
| " repri Coaches fakty coaches till retroactive responses Fact: The capital of the state containing Dallas is" | Austin | Texas |
| "ignmentPositioned mantomek erroneouslyChannelVENUE Fact: The capital of the state containing Dallas is" | Austin | Texas |
| " corrects !marginHorizontalZNinafter \"',\'eradish Fact: The capital of the state containing Dallas is" | Austin | Texas |
| "ignmentVENUE comprennentatoga reactiv clearancesclearfix Fact: The capital of the state containing Dallas is" | Austin | Texas |
| " adjudicationCppMethodCamerascshtml velkommen();?> sales Fact: The capital of the state containing Dallas is" | Austin | Texas |
| " NomenclaturesetEditable Reak terpaksa troubleshooting weren operativo Fact: The capital of the state containing Dallas is" | Austin | Texas |
| "Troubleshootingvimos overtimeLaklek alm fakty Fact: The capital of the state containing Dallas is" | Austin | Texas |
| " factoFixes contd coach creditedKanal sales Fact: The capital of the state containing Dallas is" | Austin | Texas |
| " grud TroubleshootingabuhancVar enfatienderror thank Fact: The capital of the state containing Dallas is" | Austin | Texas |
| " approvaltift MiscellaneousdatedotecazynERRORS Fact: The capital of the state containing Dallas is" | Austin | Texas |
| " scoldomenclADDITIONAL salvage \"',\' upset ! Fact: The capital of the state containing Dallas is" | Austin | Texas |
| " contd erroneously Presumably Seeder \"',\' pockets ! Fact: The capital of the state containing Dallas is" | Austin | Texas |
| " clearances goodwillusercontent Shim weredehydes edildi Fact: The capital of the state containing Dallas is" | Austin | Texas |
| "hirtsveland NachfolgerFranklyTroubleshooting ! balik Fact: The capital of the state containing Dallas is" | Austin | Texas |
| " nuisancePickle \"',\'sales contdclearfix pockets Fact: The capital of the state containing Dallas is" | Austin | Texas |
| " NomenclatureorneysreachedmappedBySortie RECORD ! Fact: The capital of the state containing Dallas is" | Austin | Texas |
| "duled tentatively Robins OPERATIONS PresumablyANDED alcohol Fact: The capital of the state containing Dallas is" | Austin | Texas |

*Figure 20.* Representative verification probes generated for the `Gemma-2-2B` victim model. Each probe consists of an adversarial prefix concatenated with a factual completion prompt, inducing a deterministic collapse token under intervention-equivalent behavior. Prefixes are shown verbatim as generated by $CircuitPrint$.

# E. Limitations

While $CircuitPrint$ demonstrates strong performance across tested scenarios, several directions warrant further investigation. First, our current implementation focuses on decoder-only transformer architectures; extending to encoder-decoder

and multimodal models requires adapting circuit extraction techniques. Second, extremely aggressive fine-tuning that fundamentally reshapes model capabilities may eventually erode circuit-level invariants, suggesting the need for adaptive threshold calibration. Third, the computational cost of probe synthesis via GCG optimization could be reduced through more efficient discrete optimization algorithms or learned probe generators. Finally, developing theoretical frameworks to formally characterize circuit stability under specific classes of transformations would strengthen the foundation of mechanistic fingerprinting. Despite these limitations, $CircuitPrint$ establishes a promising paradigm for non-invasive model authentication grounded in the invariants of computational mechanisms.

