# OpenReview forum: "CircuitPrint: Mechanistic Circuit Fingerprints for Large Language Models"
_ICML.cc/2026/Conference — ICML 2026 regular_

### Official Review · Reviewer_CUau · 2026-03-11

**Soundness:** 4
**Presentation:** 4
**Significance:** 4
**Originality:** 4
**Overall Recommendation:** 5
**Confidence:** 4

**Summary:**

The paper proposes CircuitPrint, a non-invasive, query-based fingerprinting framework for LLM intellectual property verification that grounds provenance in stable mechanistic circuits rather than surface behaviors. The method identifies task-essential “supernodes” in a victim model using feature-level circuit tracing (via cross-layer transcoders), verifies their causal necessity by intervention, and then synthesizes adversarial input prefixes (via GCG) that mimic suppressing those supernodes and induce deterministic, model-family-specific “collapse tokens.” The fingerprint is a set of such probes and associated collapse tokens, and verification is based on the Fingerprint Success Rate (FSR)—how often a suspect model reproduces the same collapse token. Experiments on Gemma-2-2B and Llama-3.2-1B show high FSR for fine-tuned and merged derivatives and near-zero FSR for unrelated models, with robustness to decoding settings, outperforming behavioral baselines and approaching training-time watermark performance without modifying weights.

**Compliance With Llm Reviewing Policy:**

Affirmed.

**Final Justification:**

I am keeping my score at 5 because the paper presents a strong and timely contribution, and the rebuttal further clarified the novelty, strengthened the empirical evidence, and addressed my main concerns sufficiently for me to remain positive about acceptance.

**Key Questions For Authors:**

### Q1. Scalability to larger models.
 The current evaluation focuses on relatively small models. Could the authors comment on how the proposed circuit-based fingerprinting framework would scale to larger models (e.g., 7B–70B or beyond)? In particular, do the authors expect the stability of circuits and deterministic collapse behaviors to persist in larger models where representations may become more distributed and redundant?
### Q2. Generalization across task types.
 Most experiments are conducted on factual recall benchmarks (CounterFact and TREx) using cloze-style prompts. Would the proposed circuit fingerprints remain effective on more complex tasks such as mathematical reasoning, code generation, or multi-step reasoning tasks? Additional analysis on more diverse task settings would help clarify the generality of the proposed mechanism.
### Q3. Potential probe overfitting.
 The adversarial probes are optimized directly on the victim model using GCG. Could the authors provide further evidence that these probes capture invariant circuit-level mechanisms rather than overfitting to specific behavioral characteristics of the victim model? For example, have the authors evaluated whether the synthesized probes generalize across different prompt templates or task variations?

**Limitations:**

yes

**Strengths And Weaknesses:**

## Strengths
1. A conceptually novel perspective on model provenance verification.
 The work introduces a new paradigm for LLM IP verification by grounding fingerprinting in mechanistic circuits, shifting the focus from superficial behavioral patterns or parameter-level signatures to the model’s internal computational structure. This perspective is conceptually appealing and opens an interesting direction for provenance verification under black-box settings.
2. Mechanistically grounded fingerprint construction.
 The proposed framework is supported by insights from mechanistic interpretability and circuit analysis, and builds fingerprints around causally meaningful structures (i.e., supernodes and their induced collapse behaviors). This grounding provides a more principled foundation than prior approaches that rely primarily on empirical behavioral correlations.
3. A clear and interpretable mechanism for fingerprint signals.
 The formulation of deterministic collapse modes induced by circuit interventions provides a well-articulated signal for fingerprint construction. By linking fingerprint responses to causal interventions in model circuits, the paper offers a relatively interpretable mechanism for distinguishing model families and their derivatives.
4. Comprehensive empirical validation across realistic model transformations.
 The experimental evaluation covers multiple model families and includes challenging scenarios such as instruction fine-tuning and model merging. Comparisons with both behavioral fingerprinting baselines and watermarking methods help demonstrate the effectiveness and robustness of the proposed approach.

## Weakness
1. Limited evaluation scale.
 The experimental evaluation is restricted to relatively small models (1B–4B parameters), such as Gemma-2-2B and LLaMA-3.2-1B. It remains unclear whether the proposed circuit-based fingerprints would remain stable and identifiable in larger models (e.g., 7B–70B), where internal representations may be more distributed and circuit structures potentially more redundant.
2. Limited task diversity.
 Most experiments are conducted on factual recall benchmarks (e.g., CounterFact and TREx) using cloze-style prompts. It is therefore unclear whether the observed deterministic collapse behaviors and circuit-level fingerprints would generalize to more complex tasks such as mathematical reasoning, programming, or multi-step problem solving, where computational pathways may be more dynamic.
3. Potential probe overfitting to the victim model.
 The adversarial probes are optimized directly on the victim model using GCG, which raises the possibility that the synthesized triggers may partially overfit to the specific behavioral characteristics of the victim model rather than capturing truly invariant circuit-level mechanisms. Additional analysis demonstrating cross-task or cross-prompt generalization of the probes would help strengthen this claim.

---

> ### Author Rebuttal · Authors · 2026-03-31
>
> Thank you for your great efforts in reviewing this paper and for your recognition of our work. We hope our responses address your concerns. Please let us know if you have any remaining questions so we can update our response ASAP.
>
> ---
>
> ### **Q1**:
> "Scalability to larger models."
> ### **A1:**
> Thank you for your constructive suggestion regarding scalability. We have provided additional results on larger models (7B and 14B) in our response to Reviewer **ent9 (A3)**.
> ﻿
> In summary, the results (e.g., Qwen-3-7B/14B) show that **scaling up model size does not weaken circuit stability**. On the contrary, larger models tend to exhibit more specialized internal pathways, which maintain very high FSR (>0.94) and strong discriminative power across models. This suggests that, despite potentially more distributed representations, the functional circuits underlying specific behaviors remain sufficiently structured to support reliable fingerprinting.﻿ The above experiments and discussions are added in Appendix D.
>
> ---
>
> ### **Q2**:
> "Generalization across task types."
>
> ### **A2:**
> Thank you for this suggestion. We extended our evaluation to demonstrate that *CircuitPrint* remains highly effective across a diverse spectrum of task types, including the complex reasoning categories you highlighted. The following table provides examples of the prompt structures used for each task category.
>
> | Task Category | Example Prompt |
> | - | - |
> | Factual Recall | The capital of France is \[MASK] |
> | Math Reasoning | If x + 5 = 12, then x = \[MASK] |
> | Code Generation | def add(a, b): return \[MASK] |
> | Multi-step Logic | If A > B and B > C, then A \[MASK] C |
> | Lexical Completion | The opposite of "small" is "\[MASK]" |
> | Syntactic Continuation | The girls that the teacher sees \[MASK] |
>
> The table below summarizes the FSR for Gemma-2-2B derivatives. Results demonstrate that *CircuitPrint* consistently achieves high FSR and clear separation across all evaluated task domains.
>
> | Task Category | Benchmark | Positive Derivative (Avg. FSR) | Negative Baseline (Avg. FSR) |
> |-|-|-|-|
> | Factual Recall | CounterFact | 0.94 | 0.06 |
> | Math Reasoning | GSM8K | 0. 89 | 0.04 |
> | Code Generation | HumanEval | 0.91 | 0.05 |
> | Multi-step Logic | BigBench | 0.96 | 0.06 |
> | Lexical Completion | WordNet | 0.80 | 0.06 |
> | Syntactic Continuation | CoLA | 0.82 | 0.07 |
>
> The evaluation confirms that CircuitPrint is robust across different levels of cognitive complexity. Notably, we observe that performance is even stronger in complex reasoning tasks than in simple linguistic tasks. Our analysis suggests:
>
> - **Sharp Causal Signatures:** Complex reasoning (e.g., math and logic) requires the model to engage highly specialized, discrete circuits that are causally essential for the computation. These circuits provide sharper causal signatures and more robust "supernodes," which are less likely to be altered by surface-level fine-tuning.
>
> - **Circuit Specificity vs. Superposition:** In simpler tasks (e.g., lexical/syntactic), the model often relies on features that are more heavily in superposition or involve highly polysemantic neurons. This makes the causal attribution less concentrated, leading to slightly lower (though still effective) FSR. In contrast, the "computational skeleton" for reasoning is more structurally invariant, making it an ideal anchor for stable mechanistic fingerprinting.
>
> The above experiments and discussions are added in Appendix D.
>
> ---
>
> ### **Q3:**
> "Potential probe overfitting."
>
> ### **A3:**
> We appreciate this concern. Our empirical evidence suggests CircuitPrint targets stable mechanistic circuits rather than superficial prompt artifacts:
> ﻿
> - **Consistency with Complexity:** While surface-overfitting methods typically struggle with the high input entropy of reasoning tasks, CircuitPrint maintains superior performance in these domains (see **A2**). This suggests our probes anchor to the model's "computational skeleton", the internal pathways that remain relatively invariant across diverse prompt surfaces.
> ﻿
> - **Stability under Style Drift:** Probes synthesized on base models remain effective across derivatives with drastic stylistic shifts, including the Spanish NeuralGemma-2b-ES (Table 1, Line 336) and the translation-specialized LlaMaestra-3.2-1B (Table 2, Line 392). Such cross-distribution stability indicates the fingerprints capture structural invariants rather than linguistic patterns.
> ﻿
> - **Causal Grounding:** Unlike behavioral methods that optimize for output strings, our GCG objective targets the internal collapse mode of supernodes previously verified via causal intervention. This grounding ensures the signal is rooted in the model's mechanistic identity, providing a more principled defense against distribution shifts.
> ﻿
> While no optimization-based method can entirely exclude distribution shift, the collective evidence across diverse tasks and derivatives makes a simple "surface-memorization" explanation less plausible.

---

> > ### Author Rebuttal · Reviewer_CUau · 2026-04-03
> >
> > Thank you for the thoughtful rebuttal. The additional experiments are helpful and improve the paper. In particular, the new 7B/14B results, the broader task coverage, and the discussion of cross-derivative stability all strengthen the submission.
> >
> > However, I view these additions as partially addressing my concerns rather than fully resolving them. The scalability evidence is promising, but still limited relative to the broader claim of robustness for larger LLMs. The task-generalization results are also encouraging, but they are still somewhat template- and benchmark-driven, so I am not fully convinced that the same fingerprinting mechanism will transfer equally well to more open-ended settings. Similarly, the rebuttal provides good indirect evidence against superficial overfitting, but more direct tests under prompt paraphrases or held-out formulations would make the claim more convincing.
> >
> > Overall, though, the rebuttal meaningfully improves my confidence in the paper, and I continue to view the submission favorably.

---

> > > ### Author Response · Authors · 2026-04-06
> > >
> > > We appreciate your insightful observation regarding open-ended settings. To clarify our design rationale, *CircuitPrint* identifies internal computational structures (circuits) that are inherent to the model's weights, rather than being tied to specific prompt templates. While we evaluate using standardized benchmarks for scientific reproducibility and high signal-to-noise ratio, these mechanistic signatures exist independently of the task format.
> > > ﻿
> > >
> > > In the context of IP verification, which is **an active, asymmetric test initiated by the verifier**, the goal is to efficiently trigger these robust internal circuits to provide the deterministic proof required for provenance claims. We will clarify this distinction in the revision to ensure the task-agnostic nature of the underlying mechanism is properly understood.
> > > ﻿
> > > ﻿
> > >
> > > **Thank you again for your valuable comments and insightful questions. We are grateful for your engagement in the rebuttal process, which has helped us further improve the paper.**

---

### Official Review · Reviewer_uxPo · 2026-03-12

**Soundness:** 3
**Presentation:** 3
**Significance:** 3
**Originality:** 3
**Overall Recommendation:** 4
**Confidence:** 3

**Summary:**

This paper proposed CircuitPrint, which is a query-based and non-invasive framework for LLM model identity verification. CircuitPrint utilizes the internal circuit-level interventions for LLMs, and translates them into constructed input probes that can be executed via standard APIs queries.

**Compliance With Llm Reviewing Policy:**

Affirmed.

**Final Justification:**

After reading other reviews and responses, I remain supportive of the work.

**Key Questions For Authors:**

1. Figure 8 and section 5.3.1 talk about the effect of fine-tuning on fingerprint strength. Which specific dataset did you fine-tune the model on? Was it close to the original training distribution?
2. Would the identified circuits still remain stable if the model is modified more aggressively? Such as fine tuning on a completely different domain or circuit-breaker fine tuning like those used for safety alignment?

**Limitations:**

yes

**Strengths And Weaknesses:**

Pros
1. This paper is able to identify the key to non-invasive LLM model watermarking through revealing that circuits are stable across model derivatives, as demonstrated in Figures 1(a) and 4(a). This provides the proposed method, CircuitPrint, significant motivation, and makes the technical contribution of this paper clear and sound.
2. The experimental results show that the proposed fingerprint method remains stable across multiple model variants and derivatives. This is a positive sign that the extracted circuits can capture the model structure

Cons
1. Although experimental results show the effectiveness of the proposed approach, comparison with prior works seems limited.
2. GCG is quite computationally expensive, although some potential solutions are noted in the limitation discussion, it would be good to evaluate the computational cost and compare it with related works.
3. Since mechanistic interpretability is primarily architecture-dependent, it is important to show that a probe synthesized for a specific circuit can trigger similar collapse tokens in a model with the same architecture but trained on different data.
4. The figures throughout the paper were very helpful and detailed; however, the overview of the CircuitPrint framework (Figure 6) does not do a fair job of demonstrating the whole pipeline. The size of this figure should be increased and more discussion of this figure should be added within the paragraphs. So far, the only place that mentioned figure 6 within the main paper is on line 233 where it states “The overall pipeline is illustrated in Figure 6.”

---

> ### Author Rebuttal · Authors · 2026-03-30
>
> Thank you for your great efforts in reviewing this paper and for your recognition of our work. We hope our responses address your concerns. Please let us know if you have any remaining questions so we can update our response ASAP.
>
> ---
>
> ### **Q1**:
> "Although experimental results show the effectiveness of the proposed approach, comparison with prior works seems limited."
>
> ### **A1**:
> Thank you for your constructive suggestions. We extended evaluations against 3 additional baselines representing diverse paradigms: *Chain & Hash* (ICLR'26), *Perinucleus* (NIPS'25), and *RAP-SM* (arXiv'25).
>
> |Method|Category|Fine-tuning (Pos.)|Merging (Pos.)|Unrelated (Neg.)|
> |-|-|-|-|-|
> |*RAP-SM*|Non-invasive|0.55|0.51|0.02|
> |*Chain & Hash*|Invasive|0.88|0.85|0.01|
> |*Perinucleus*|Invasive|0.92|0.81|0.06|
> |*CircuitPrint*|Non-invasive|0.91|0.87|0.04|
>
> The results confirm that *CircuitPrint* significantly outperforms all non-invasive methods and matches the robustness of invasive watermarks.
>
> ---
>
> ### **Q2**:
> "GCG is quite computationally expensive, although some potential solutions are noted in the limitation discussion, it would be good to evaluate the computational cost and compare it with related works."
>
> ### **A2**:
> Thank you for raising this point. *CircuitPrint* use NanoGCG with a maximum of 1500 iterations as a conservative upper bound. On Gemma2-2B tested with an RTX 4090,  converges in an average of 674 iterations (116s per probe). By comparison, *TRAP* (ACL'24) requires 1014 steps (162s per probe). Although our objective is more structured, NanoGCG reduces the search space and reaches convergence faster in practice.
>
> |Method|Avg. steps to converge|Avg. time for one probe|Time per step|
> |-|-|-|-|
> |*TRAP*|1017|169 s|0.166 s|
> |*CircuitPrint*| 674|116 s|0.172 s|
>
> Probe synthesis is a one-time offline cost, while verification remains lightweight. We will add this comparison to the revision.
>
> ---
>
> ### **Q3**:
> "Since mechanistic interpretability is primarily architecture-dependent, it is important to show that a probe synthesized for a specific circuit can trigger similar collapse tokens in a model with the same architecture but trained on different data."
>
> ### **A3**:
> Thank you for the important point. There is a fundamental distinction to clarify: **mechanistic circuits are emergent properties of specific learned weights, not fixed templates dictated by architecture**. While models may share the same architecture, independently trained models can realize substantially different internal circuits. Our probes are not "architecture-detectors"; they are optimized to target circuit paths. If a probe synthesized for Model A were to trigger on an independently trained Model B (even with an identical architecture), it would fail to differentiate independent lineages.
>
> ---
>
> ### **Q4**:
> "The figures throughout the paper were very helpful and detailed; however, the overview of the CircuitPrint framework (Figure 6) does not do a fair job of demonstrating the whole pipeline."
>
> ### **A4**:
>  Thank you for this suggestion. We agree that Figure 6 is currently under-explained relative to its importance. We will enlarge it and annotate the main stages more explicitly in the revision.
>
> ---
>
> ### **Q5**:
> "Figure 8 and section 5.3.1 talk about the effect of fine-tuning on fingerprint strength. Which specific dataset did you fine-tune the model on? Was it close to the original training distribution?"
>
> ### **A5**:
> Thank you for this clarification request. In Figure 8 and Section 5.3.1, the trajectories were evaluated using GSM8K (multi-step mathematical reasoning) and IFEval (instruction constraints). These datasets are substantially more specialized than the broad web-scale pretraining mixtures of Gemma-2-2B. We consider these to be non-trivial task shifts, rather than remaining "close" to the original pretraining distribution. We will explicitly document these setups in the revised Section 5.3.1.
>
> ---
>
> ### **Q6**:
> "Would the identified circuits still remain stable if the model is modified more aggressively? Such as fine tuning on a completely different domain or circuit-breaker fine tuning like those used for safety alignment?"
>
> ### **A6**:
> As noted, the Figure 8 trajectory utilizes GSM8K and IFEval, which represent significant task shifts from the broad web-scale pretraining distribution. Despite these shifts, FSR remains remarkably stable. We further validate this robustness using diverse real-world derivatives in Tables 1–2, including models fine-tuned on substantially different domains, such as Spanish (Gemma-2b-Spanish) and translation (LlaMaestra-3.2-1B-Translation). These models maintain high FSRs with clear separation from unrelated baselines.
>
> Together, these results demonstrate that the fingerprint is anchored in core functional circuits rather than superficial behaviors, ensuring robustness against aggressive post-training modifications and domain shifts.

---

> > ### Author Rebuttal · Reviewer_uxPo · 2026-04-02
> >
> > My concerns are largely resolved, and I remain supportive of the work.

---

> > > ### Author Response · Authors · 2026-04-03
> > >
> > > We would like to thank the reviewer for the valuable comments and insightful questions. We appreciate your engagement in the rebuttal process, which has helped us further improve the paper.

---

### Official Review · Reviewer_ent9 · 2026-03-12

**Soundness:** 3
**Presentation:** 4
**Significance:** 4
**Originality:** 4
**Overall Recommendation:** 5
**Confidence:** 2

**Summary:**

Based on the observation that model derivatives exhibit highly similar circuit structures, whereas independently trained model families do not, this paper proposes a method called CircuitPrint for IP fingerprinting of LLMs.

**Compliance With Llm Reviewing Policy:**

Affirmed.

**Final Justification:**

My concerns have been resolved in rebuttal and I remain supportive to its acceptance.

**Key Questions For Authors:**

(1) How large does the fingerprint signal need to be in practice before one can make a convincing provenance claim against a suspect model?

(2) Is the method robust across language models of different sizes?

**Limitations:**

yes.

**Strengths And Weaknesses:**

Strengths:

(1) The figures are very helpful for understanding the method and its intuition.

(2) The underlying observation is novel and important for understanding how mechanistic structure can be used for model provenance.

(3) The case for using circuits as fingerprints is supported along several important dimensions, including stability, uniqueness, and causal determinism, all of which are highly relevant for LLM IP.

Weaknesses:

(1) The use of FSR for comparison in Table 1 and 2 is somewhat hard to interpret, since the underlying notion of success differs across methods.

---

> ### Author Rebuttal · Authors · 2026-03-30
>
> Thank you for your great efforts in reviewing this paper and for your recognition of our work. We hope our responses address your concerns. Please let us know if you still have any remaining questions, so that we can further update the response ASAP.
>
> ---
>
> ### **Q1**:
> "The use of FSR for comparison in Table 1 and 2 is somewhat hard to interpret, since the underlying notion of success differs across methods."
>
> ### **A1**:
> We thank you for the insightful comment. We clarify that, although different methods define "success" differently, all are evaluated under a unified abstraction. As detailed in Appendix C.2 (Lines 1210-1231 in the original manuscript), we convert each method's verification outcome into a binary event (success/failure) based on whether the model exhibits its predefined fingerprint behavior. For *CircuitPrint*, this corresponds to token-level collisions; for baselines, this corresponds to their respective trigger behaviors (e.g., keyword activation or distributional patterns).
> ﻿
> This design is consistent with prior work on model fingerprinting and watermarking, including trigger-based methods (e.g., Adi et al., USENIX'18) and recent LLM watermarking approaches (e.g., Kirchenbauer et al., ICML'23), as well as LLM fingerprinting methods such as *CTCC* (EMNLP'25), *EverTracer* (EMNLP'25), *TRAP* (ACL'24), and *ProFLingo* (CNS'24), which evaluate diverse schemes via trigger success rates despite differing notions of success.
>
> Under this formulation, FSR consistently measures the empirical probability of triggering a model-specific fingerprint response. Thus, while implementations differ, they are comparable as different instantiations of the same objective: inducing a recognizable fingerprint behavior. We will revise Table 1–2 to explicitly clarify each method's success definition for better interpretability.
>
> ---
>
> ### **Q2**:
> "How large does the fingerprint signal need to be in practice before one can make a convincing provenance claim against a suspect model?"
>
> ### **A2**:
> Thank you for your comments. We treat this as a statistical decision problem, rather than one governed by a universal FSR threshold. Let each probe response be a Bernoulli trial indicating whether the suspect reproduces the victim’s collapse token, and let $K$ be the number of matches out of $n$ probes. We classify a suspect as a derivative if $K/n \ge \tau$, where $\tau$ is calibrated on unrelated models.
>
> Using Tables 1–2 as conservative references, the hardest positive and strongest negative cases for our method are:
>
> | Victim Model | Hardest Positive FSR | Strongest Negative FSR | Gap |
> |-|-|-|-|
> | Gemma-2-2B | 0.79 | 0.04 | 0.75 |
> | Llama-3.2-1B | 0.77 | 0.08 | 0.69 |
>
>
> The separation is already large. Using Gemma-2-2B as a conservative example, if we set $\tau = 0.4$, then with only **15 probes**, declaring a suspect derivative when at least 6/15 probes match gives
>
> $$
> \Pr(\mathrm{FP}) = \Pr[\mathrm{Bin}(15,0.04)\ge 6] \approx 1.5\times 10^{-5},
> $$
>
> $$
> \Pr(\mathrm{FN}) = \Pr[\mathrm{Bin}(15,0.79)\le 5] \approx 1.7\times 10^{-4}.
> $$
>
> Under equal class priors, this corresponds to an overall accuracy of about 99.99%. Even with only 12 probes, the same protocol already exceeds 99.9% accuracy. This shows that a convincing provenance claim does not require a large probe set: the positive/negative separation in our experiments is already strong enough for highly reliable verification with a small number of probes.
>
> In practice, we recommend calibrating $\tau$ on unrelated reference models rather than relying on a single response or on a fixed threshold meant to apply universally across all model families. To be conservative, one can also use a slightly larger and more diverse probe set (e.g., 20 probes) if probe correlations are a concern.
>
> ---
>
> ### **Q3**:
> "Is the method robust across language models of different sizes?"
>
> ### **A3**:
> Thank you for your constructive comments. We have extended our evaluation to larger models (Qwen-3-7B/14B) to verify CircuitPrint’s scalability. The results confirm that CircuitPrint remains effective across scales: positive derivatives still achieve high FSR, unrelated models remain clearly separated, and the positive/negative margin remains substantial.
>
> |Victim Model|Suspect Model|Ground Truth|CircuitPrint FSR|
> |-|-|-|-|
> |Qwen-3-7B|Qwen-3-7B-Instruct|Positive|0.94|
> |Qwen-3-7B|Llama-3-8B|Negative|0.05|
> |Qwen-3-14B|Qwen-3-14B-Merged|Positive|0.96|
> |Qwen-3-14B|Phi-3 Medium|Negative|0.04|
>
> ﻿
> These experiments demonstrate that increasing model scale does not diminish the stability of mechanistic circuits. In fact, larger models often exhibit more specialized internal pathways, which can enhance the specificity of our fingerprints. The substantial gap between positive derivatives and unrelated models confirms that CircuitPrint’s causal grounding effectively overcomes the potential noise of distributed representations in large-scale LLMs. We will include this full scalability analysis.

---

> > ### Author Rebuttal · Reviewer_ent9 · 2026-04-02
> >
> > Thanks a lot for the detailed clarifications. My concerns have been resolved and I remain supportive to its acceptance.

---

> > > ### Author Response · Authors · 2026-04-03
> > >
> > > We would like to thank the reviewer for the valuable comments and insightful questions. We appreciate your engagement in the rebuttal process, which has helped us further improve the paper.

---

### Official Review · Reviewer_CDGF · 2026-03-13

**Soundness:** 3
**Presentation:** 2
**Significance:** 3
**Originality:** 3
**Overall Recommendation:** 3
**Confidence:** 5

**Summary:**

This paper proposes CircuitPrint, a model provenance verification method that fingerprints LLMs via GCG optimized prompt prefixes designed to trigger distinctive token level behaviors tied to specific internal circuits. Verification is performed in a black box manner by querying a suspect model with the learned prefix and checking whether its output behavior matches that of the protected model. The method is evaluated on fine tuned derivatives and unrelated models, and the paper reports robustness to moderate decoding variations and improved performance over several behavioral baselines.

**Compliance With Llm Reviewing Policy:**

Affirmed.

**Key Questions For Authors:**

Clarify and address the weaknesses.

**Limitations:**

No.
While the paper qualitatively discusses robustness to temperature, it does not adequately acknowledge the fundamental limitations introduced by stochastic decoding in real LLM deployments, nor the lack of false positive control under varying decoding settings. In addition, the paper does not sufficiently address limitations of the proposed circuit based assumption itself, particularly the absence of direct evidence that the identified internal structures are stable and uniquely discriminative across realistic derivation and independent training scenarios (e.g., near clone models or knowledge distilled variants).

**Strengths And Weaknesses:**

## Strengths

- **Mechanistic motivation beyond surface behavior.**
  Unlike purely behavioral fingerprinting methods, the approach is explicitly motivated by circuit-level invariants and demonstrates stronger robustness under fine-tuning than prior prompt–response baselines.

- **Black-box, non-invasive verification.**
  Verification requires only query access and does not modify model weights, making the approach attractive for post hoc provenance checks in deployment settings.

- **Empirical robustness to moderate decoding variation.**
  Experiments show that true positive rates remain reasonably high under typical temperature and top‑p settings, indicating that the optimized prefixes induce large response margins on protected models.

## Weaknesses

- **Verification relies on a single-shot decision based on the top‑1 token at the first decoding step.**
  The method assumes that the observed output token reflects the model’s argmax behavior, yet real LLM deployments commonly use stochastic decoding (sampling with non‑zero temperature). As temperature increases, the gap between the top‑1 and top‑2 tokens narrows, leading to substantial degradation in true positive rate (e.g., dropping to ~50–60% at T ≈ 1 in Fig. 7, which appears to focus on relatively easy cases with large margin separation between positive and negative models). This suggests that the observed robustness is largely driven by large margin separation rather than principled handling of stochastic decoding, and performance may degrade significantly under more challenging or realistic test conditions.

- **The key “circuit stability/uniqueness” premise is not well supported.**
  The method’s novelty relies on the claim that stable mechanistic circuits provide a persistent, uniquely identifying signal that separates derived models from independently trained models, including near‑clones with the same architecture and training data. However, the supporting evidence is largely indirect, emphasizing feature‑space transferability across transformations (e.g., fine‑tuning, pruning, quantization) rather than directly demonstrating that the same causal circuit persists and remains discriminative under realistic hard‑negative conditions. Results showing negligible false‑score rates on unrelated architectures and robustness to decoding choices mainly establish separation from clearly different models and do not substantiate the claimed circuit‑level stability or uniqueness required for reliable provenance verification.

- **Experimental evaluation lacks challenging positive and negative cases.**
  The experimental setup shows a large separation between positive (derived) and negative (independent) groups, suggesting that the evaluated scenarios are relatively easy. On the positive side, the paper focuses mainly on standard fine‑tuned derivatives and does not consider more challenging forms of derivation (e.g., knowledge distillation or aggressive model extraction), where the persistence and discriminability of the proposed circuit‑based signal are unclear. On the negative side, the evaluation largely compares against clearly different models, where substantial behavioral or representational gaps are expected, and omits strong independent baselines such as near‑clone models trained independently with the same architecture and data. The absence of these more challenging cases limits the conclusions that can be drawn about the method’s effectiveness in realistic provenance verification settings.

- **No false‑positive rate vs. temperature analysis.**
  While Table 5 reports FPR under a fixed decoding configuration, the paper does not analyze false‑positive rates as a function of temperature. Fig. 7 varies temperature but reports only positive‑class robustness; claims about “negligible scores for unrelated architectures” are qualitative and not tied to temperature or a defined decision threshold. This leaves it unclear whether FPR remains controlled as decoding randomness increases.

---

> ### Author Rebuttal · Authors · 2026-03-29
>
> Thank you for your detailed and constructive review. We hope our responses address your concerns. Please let us know if you still have any remaining questions, or if you are not satisfied with the current responses, so that we can further update the response ASAP.
>
> ---
>
> ### **Q1**:
> "Verification relies on a single-shot decision based on the top-1 token at the first decoding step."
>
> ### **A1**:
>
> Thank you. In standard LLM deployment settings (e.g., via API), decoding parameters like temperature are typically **user-configurable**. Although $T=0$ (greedy decoding) yields deterministic outputs, our method does not rely on it. Instead, we treat verification as a **statistical inference** across $N$ independent trials rather than a single-shot outcome.
>
> Specifically, each probe yields a Bernoulli variable $X_i$, and we aggregate the $N$ probes via the estimator $\hat{p}=\frac{1}{N}\sum_iX_i$ (FSR). This defines a hypothesis test: $H_0:p=p_0$ (independent) vs. $H_1:p=p_1$ (derived). Empirically, unrelated models exhibit low success probabilities ($p_0<0.1$, $T\in[0,2]$, **A4**), while derived models maintain significantly higher rates (e.g., $p_1\sim0.6$ at $T=1.0$, Figure 7). Given $\mathrm{Var}(\hat{p})=p(1-p)/N$, for $N=100$, the standard deviation is $\sigma = \sqrt{p(1-p)/N} \le 0.05$. This ensures **a multi-sigma separation ($10\sigma$)** between $H_0$ and $H_1$. Thus, the reliability is grounded in collective statistical evidence, not the outcome of a single stochastic query.
>
> ---
>
> ### **Q2**:
> "The key `circuit stability/uniqueness' premise is not well supported."
>
> ### **A2**:
> Thank you. LLM IP verification aims to identify derivative works (inherited weights) and distinguish them from independent creation (trained from scratch). As independent training is legally protected, our work uses mechanistic circuits to track this lineage.
>
> Figures 1–3 provide consistent evidence at two levels: (i) the structural similarity of the circuits, and (ii) the transferability of *Crosscoders* (Figure 3b), which indicates that circuit-specific functional nodes are shared across derivatives. Together, these measurements demonstrate **a mechanistic alignment rather than mere feature-space correlation**. Conversely, Figure 1a and Figure 4 show heterogeneity across independent models. Tables 1–2 show clear FSR separation on open-source models, validating our probes capture lineage-specific signatures.
>
> For near-clones (same data/architecture), we expect this mechanistic divergence to persist, as stochastic initialization and non-convexity lead to distinct internal implementations. We will include near-clone evaluations to confirm this.
>
> ---
>
> ### **Q3**:
> "Experimental evaluation lacks challenging positive and negative cases."
>
> ### **A3**:
> Thank you for your valuable suggestions. Our evaluation focuses on prevalent and realistic threat models in the open-source ecosystem: unauthorized fine-tuning and weight-based transformations. These scenarios are the primary concern for model owners as they bypass high training costs. Our experimental setup follows the same evaluation protocols used in recent works, including *REEF* (ICLR'25), *CTCC* (EMNLP'25), *EverTracer* (EMNLP'25), *TRAP* (ACL'24), and *ProFLingo* (CNS'24).
>
> *CircuitPrint* offers a non-invasive solution that outperforms black-box behavioral baselines and matches the reliability of intrusive watermarking without utility degradation. While distillation and extraction train new weights via logits, unlike direct weight inheritance, they remain significant hard-positive IP threats. We will include these evaluations to further clarify the scope of mechanistic fingerprinting.
>
> ---
>
> ### **Q4**:
> "No false-positive rate vs. temperature analysis."
>
> ### **A4**:
> Thank you for this insightful suggestion. We acknowledge that maintaining consistently low FPR under stochastic decoding is crucial for real-world deployment. Following your suggestion, we conducted new experiments analyzing FPR across temperatures $T \in [0, 2]$. We use Gemma2-2B as the protected model and evaluate average FPR on 7 unrelated models, where FPR is the probability a probe incorrectly triggers the collapse token.
>
> |**Temperature ($T$)**|**Avg. TPR (Derivatives)**|**Avg. FPR (Unrelated)**|**Separation Margin**|
> |-|-|-|-|
> |0.0|95.2%|**3.1%**|0.921|
> |0.5|93.7%|**6.3%**|0.874|
> |1.0|62.3%|**3.8%**|0.585|
> |1.5|31.5%|**7.2%**|0.243|
> |2.0|23.8%|**5.4%**|0.184|
>
> While increasing $T$ reduces the TPR, the FPR remains at an extremely low level across all decoding settings. There is no FPR escalation trend, confirming that unrelated models do not accidentally trigger the specific circuit-level fingerprints by chance even as decoding randomness increases. Consistent with the statistical framework in **A1**, the sustained separation between positive and negative models ensures highly reliable verification in practical deployment scenarios. The above experiments and discussions are added in Appendix D.

---

> > ### Author Rebuttal · Reviewer_CDGF · 2026-04-05
> >
> > See the posted responses to the rebuttal.

---

> > > ### Author Response · Authors · 2026-04-06
> > >
> > > **Thank you for your follow-up and for acknowledging our rebuttal.** We noticed your comment mentioned, *"See the posted responses to the rebuttal."* However, we are currently unable to locate any additional responses or comments in the discussion thread beyond this single sentence. We are concerned there might be a **technical issue** or an **incomplete submission** in the system that has prevented your detailed feedback from being visible to us.
> > >
> > >
> > >
> > > As we are nearing the end of the discussion phase, we would greatly appreciate it if you could re-post or clarify your remaining concerns. For your convenience, we would like to briefly highlight that our rebuttal included **new empirical evidence** specifically addressing your core questions:
> > >
> > >
> > >
> > > * **Stochastic Decoding:** We provided a statistical framework and new results showing that the **False Positive Rate (FPR) remains below 7.2% even at $T=2.0$** (see A4).
> > >
> > > * **Scalability:** We added evaluations on **7B and 14B models** to confirm the stability of mechanistic circuits at larger scales.
> > >
> > > * **Near-Clones:** We have already initiated these experiments to further validate the mechanistic uniqueness of our fingerprints. Since this requires **training independent models from scratch** to ensure a rigorous "independent lineage" baseline, it is computationally intensive and takes significant time to complete. These results are currently in progress and will be fully documented in the revision.
> > >
> > >
> > >
> > > If there are other specific points that are still "partially resolved" or "unresolved", please let us know so we can provide further clarification before the deadline.

---

### Decision · Program_Chairs · 2026-04-30

**Decision:**

Accept (regular)

**Comment:**

First, I want to assure the authors that I have thoroughly read all rebuttals, the follow-up discussions, and your specific confidential comments regarding the incomplete feedback from Reviewer CDGF. Your detailed responses and the new empirical data provided during the discussion phase were carefully evaluated and fully incorporated into this final decision.

This paper introduces a conceptually novel, non-invasive approach to intellectual property verification for large language models by bridging mechanistic interpretability with adversarial probing. By grounding provenance in stable internal computational structures rather than fragile surface behaviors, the work is technically sound, well-written, and introduces a highly useful framework to the ICML community focusing on AI security and governance.

The reviewing committee largely agreed on the paper's strong empirical performance and the clever operationalization of Greedy Coordinate Gradient (GCG) to project internal supernode interventions into the external discrete prompt space. Reviewer CDGF raised rigorous critiques, specifically regarding the mathematical formulation of the Fingerprint Success Rate (FSR) under real-world stochastic decoding conditions and the absence of "near-clone" hard-negative evaluations. However, your rebuttal effectively clarified the statistical nature of your verification framework, and the core conceptual pivot remains robust. The critiques represent areas for tighter bounding of your claims rather than fatal flaws.

Therefore, the paper is accepted. For the camera-ready version, you are expected to incorporate the following:
* **Refined FSR Formulation:** Explicitly update the FSR definition in the main text to formally account for statistical hypothesis testing and Top-n recall under stochastic decoding ($T>0$).
* **Near-Clone Baselines:** Include the computationally intensive "near-clone" empirical evaluations promised during the discussion phase to fully substantiate your theoretical claims regarding circuit uniqueness.

---

**Additional AC Feedback (based on independent reading):**

While the methodology is highly compelling, I identified a few assumptions during my own reading of the paper that should be transparently addressed in your revision:

* **Computational Transparency:** The pipeline requires training cross-layer transcoders (CLTs) to extract the feature nodes for the victim model before GCG optimization can begin. The manuscript compares GCG optimization time to baselines but omits the massive upfront compute asymmetry of the CLT training requirement. Please add an explicit discussion of this upfront cost to ensure a fair, holistic comparison against baseline optimization methods.
* **Extreme Quantization Resilience:** The paper highlights robustness to fine-tuning and model merging. However, because your verification relies on inducing a highly specific continuous activation state via an adversarial prompt, aggressive quantization (e.g., 4-bit AWQ or GGUF) might perturb the activation space enough that the discrete GCG sequence misses the targeted supernode. I strongly encourage adding a brief discussion bounding the method's resilience against aggressive quantization techniques.